

# In-situ estimation of subsurface hydro-geomechanical properties using the groundwater response to Earth and atmospheric tides

Timothy C. McMillan[1,2], Martin S. Andersen[1], Wendy A. Timms[3], and Gabriel C. Rau[1,4]

[1]School of Civil and Environmental Engineering, The University of New South Wales, Sydney, Australia
[2]School of Mineral and Energy Resource Engineering, The University of New South Wales, Sydney, Australia
[3]School of Engineering, Deakin University, Waurn Ponds, Australia
[4]Institute of Applied Geosciences (AGW), Karlsruhe Institute of Technology (KIT), Karlsruhe, Germany

**Correspondence:** Gabriel C. Rau (gabriel.rau@kit.edu)

**Abstract.** Subsurface hydro-geomechanical properties crucially underpin the management of Earth's resources, yet they are predominantly measured on core-samples in the laboratory while little is known about the representativeness of in-situ conditions. The impact of Earth and atmospheric tides on borehole water levels are ubiquitous and can be used to characterise the subsurface. We illustrate that disentangling the groundwater response to Earth and atmospheric tidal forces in conjunction
with hydraulic and linear poroelastic theories leads to a complete determination of the whole hydro-geomechanical parameter space for unconsolidated systems. Further, the characterisation of consolidated systems is possible when using literature estimates of the grain compressibility. While previous field investigations have assumed a Poisson's ratio from literature values, our new approach allows for its estimation under in-situ field conditions. We apply this method to water level and barometric pressure records from four field sites with contrasting hydrogeology. Estimated hydro-geomechanical properties (e.g. spe-
cific storage, hydraulic conductivity, porosity, shear-, Young's- and bulk- moduli, Skempton's and Biot-Willis coefficients and undrained/drained Poisson's ratios) are comparable to values reported in the literature, except for consistently negative drained Poisson's ratios which are surprising. Our results reveal an anisotropic response to strain, which is expected for a heterogeneous (layered) lithological profile. Closer analysis reveals that negative Poisson's ratios can be explained by differing in-situ conditions to those from typical laboratory core tests and the small strains generated by Earth and atmospheric tides. Our new
approach can be used to passively, and therefore cost-effectively, estimate subsurface hydro-geomechanical properties representative of in-situ conditions. Our method can be used to improve our understanding of the relationship between geological heterogeneity and geomechanical behaviour.

## 1 Introduction

A perpetual challenge for subsurface water, mineral resource or geotechnical projects is a proper characterisation of the physical properties that may have bearings on the rate of resource extraction, operation, safety and environmental impact of the



project. The main reason for this challenge is the subsurface's heterogeneous nature and that the sampling density necessary to describe it may be prohibitively expensive (e.g. by drilling and testing of core). This issue is further exacerbated by the difficulty in approximating in-situ environments in laboratory testing in regards to both scale and subsurface pressures (Hoek and

Diederichs, 2006; Cundall et al., 2008; Bouzalakos et al., 2016). These difficulties may be overcome by in-situ characterisation of hydro-geomechanical properties of the subsurface (Villeneuve et al., 2018). Here, the in-situ pressure, stress conditions, and the scaling and inclusion of heterogeneities can achieve a more representative estimate than possible from selective laboratory testing.

    Detailed time-series analysis of water table fluctuations in boreholes due to Earth and atmospheric tides (EAT) has been

shown to be capable of deriving hydro-geomechanical properties (Hsieh et al., 1987; Rojstaczer and Agnew, 1989; Zhang et al., 2019). Further, with the assumption of key variables, previous authors have also been able to extend the use of EAT to estimate additional subsurface hydro-geomechanical properties (Bredehoeft, 1967; Beavan et al., 1991; Cutillo and Bredehoeft, 2011). However, methods which use EAT, referred to as tidal subsurface analysis (TSA) techniques, remains underutilised.

    EAT are natural phenomena, causes by celestial bodies (e.g., Sun or Moon), that occur throughout the Earth's crust, which

have been measured and analysed in the subsurface since the mid-$20^{th}$ century (McMillan et al., 2019). Traditionally these techniques have been focused on either Earth tides (Bredehoeft, 1967; Hsieh et al., 1987; Cutillo and Bredehoeft, 2011; Zhang et al., 2019; Burbey, 2010), barometric pressure (Clark, 1967; Cutillo and Bredehoeft, 2011) or atmospheric tide loading (Acworth et al., 2016; McMillan et al., 2019; Rau et al., 2020) of the confined subsurface. Bredehoeft (1967) first proposed that once specific storage is calculated from the groundwater response to Earth tides, an aquifer porosity and compressibility

can be determined from the formation pressure response to a uniformly distributed surface load such as caused by barometric pressure changes (Narasimhan et al., 1984; Rojstaczer, 1988; Rojstaczer and Riley, 1990; Ritzi et al., 1991; Burbey et al., 2012). This concept has been reiterated in the literature but, to the best of our knowledge, never solved without the use of either an assumed Poisson's ratio or bulk modulus (Cutillo and Bredehoeft, 2011) due to difficulties in attributing the superimposed EAT effects to their appropriate drivers (e.g. celestial body gravitational or atmospheric loading forces). Recent work estimating

amplitudes and phases using *harmonic least squares* (HALS) and synthetically predicted ETs has demonstrated that separating tidal components of very similar frequencies is now possible (Rau et al., 2020). This has opened opportunities to revisit existing methods to create a new integrated approach that remove the need to assume moduli and therefore instead derive the variables from the data.

    In this paper, the theory of the groundwater response to Earth and atmospheric tides is combined, thereby providing a new

methodology for the estimation of the primary subsurface hydrogeomechanical properties (storage, hydraulic conductivity, poroelastic properties). This new method improves upon the work of Cutillo and Bredehoeft (2011), as it quantitatively disentangles the groundwater response to Earth and atmospheric tides within the frequency domain, removing non-harmonic signals (e.g. precipitation and episodic recharge events) and allowing the separate and objective estimation of properties from each driver before combining the strain responses. Here, the hydraulic and linear poroelastic works of Hsieh et al. (1987),

Rojstaczer and Agnew (1989), Beavan et al. (1991) and Rau et al. (2020) are integrated and combined, leading to a complete determination of the parameter space for unconsolidated systems. Further, the characterisation of consolidated systems is pos-





| Tidal component (Darwinian name) | Frequency $(cpd)$ | Tidal potential $(m^2\,s^{-2})$ | Tidal gravity variation $(m\,s^{-2})$ | Tidal dilatation / areal strain $(-)$ | Description | Attribution |
|---|---|---|---|---|---|---|
| $M_2$ | 1.932274 | 42.060943 | $6.477 \cdot 10^{-5}$ | $2.625 \cdot 10^{-7}$ | Principal lunar semi-diurnal | Earth |
| $S_2$ | 2.000000 | 19.309855 | $2.973 \cdot 10^{-5}$ | $1.205 \cdot 10^{-7}$ | Principal solar semi-diurnal | Atmosphere/Earth |

**Table 1.** Table of $M_2$ and $S_2$ tidal components, tidal potential, gravity and dilatation using tidal predictions (this does not include local variations). Extracted from Agnew (2010) and McMillan et al. (2019).

sible when using literature estimates of the grain compressibility (van der Kamp and Gale, 1983; Green and Wang, 1990). Finally, the new methodology is applied to groundwater and atmospheric pressure records in five boreholes from four sites to estimate hydrogeological and geomechanical properties of various consolidated and unconsolidated stratigraphies.

## 2 Theoretical background

### 2.1 Extracting tidal components

Atmospheric heating and the gravitational pull of celestial bodies exert a loading of the Earth's crust (Agnew, 2010). The gravity variations and loading exerted by the movement of these celestial bodies (i.e., the Moon and Sun), as shown in Table 1, cause stress and strain responses in the Earth's crust. This causes a subsurface strain signal that is composed of numerous superimposed signals of various frequencies and amplitudes. For undrained conditions (pressurized) of either confined or

semi-confined aquifers, this strain manifests as a groundwater pore pressure fluctuation (McMillan et al., 2019). A conceptual illustration of these processes is shown in Figure 1.

Three variables are required to calculate subsurface properties using specific harmonic components (McMillan et al., 2019): (1) a computed dilatation strain due to Earth tides (denoted by the superscript $ET$); (2) measured barometric pressure (denoted

by the superscript $AT$ in later equations); and (3) measured groundwater heads (denoted by the superscript $GW$). First, a moving average spanning across a time period of 3 days is applied. The tidally induced frequency components are then extracted by using *harmonic least-squares* (HALS) to estimate the harmonic components caused by tides whose frequencies are well known (Hsieh et al., 1987; Xue et al., 2016; Rau et al., 2020; Schweizer et al., 2021).The moving average acts like a high-pass filter and the extraction of the tidal components at specific frequencies means that longer frequency and episodic events

(atmospheric pressure changes related to weather systems passing across a site, rainfall and recharge events, and recession from pumping or droughts) present in the groundwater level/pressure data is discarded and therefore of no consequence for the analysis. Further, since our analysis is exclusively valid for the semi-confined and confined subsurface, water movement in the overlying shallow vadose zone and unconfined saturated zone is irrelevant. The extracted frequency components are



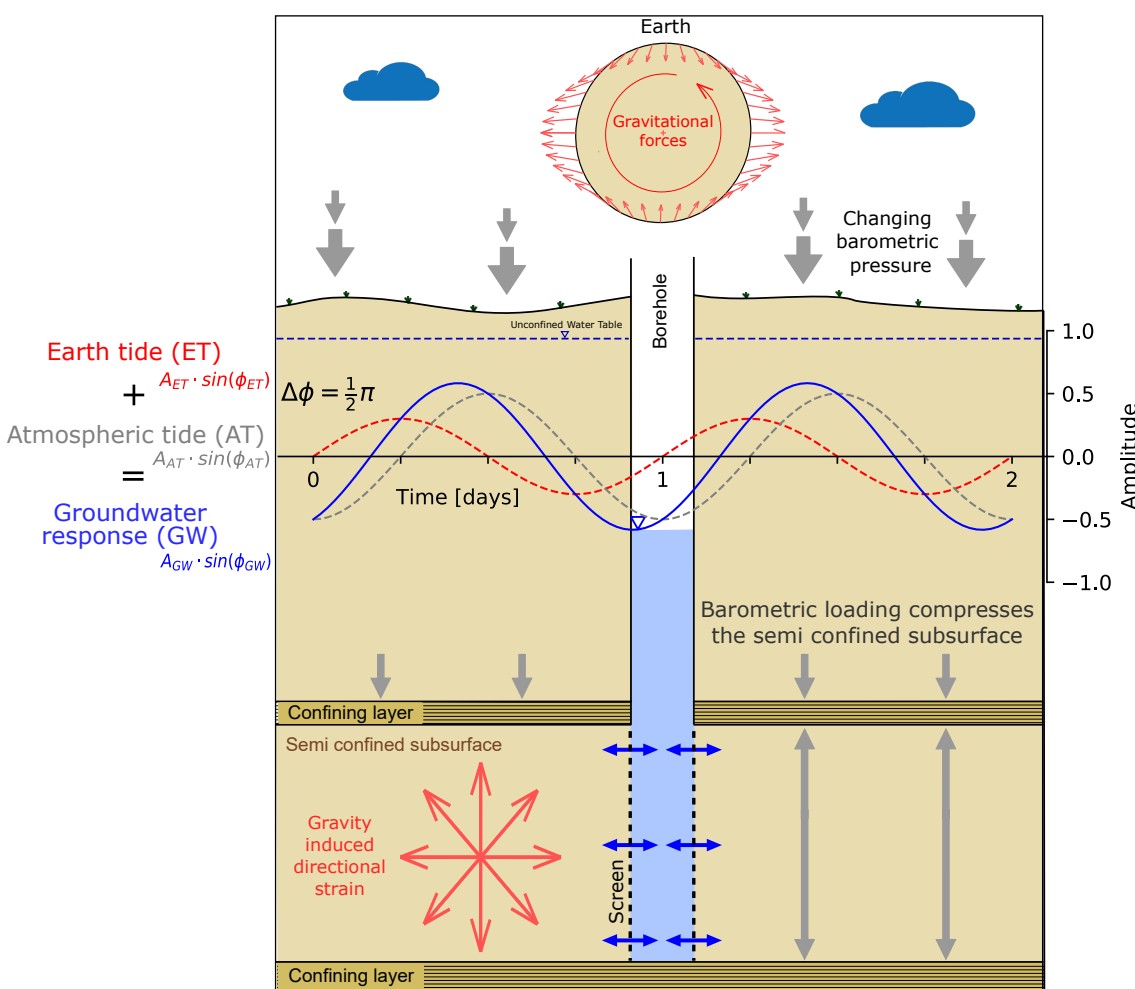

**Figure 1.** Representation of groundwater pressure head measured in a well penetrating a confined aquifer with a relatively rigid matrix subjected to ET (red) and AT (grey), adapted from (McMillan et al., 2019). The result of these two effects can be expressed as a function of harmonic addition within the groundwater level. Here, the gravity-induced directional strain and vertical barometric loading/unloading combine to force water into and out of the well.





complex numbers at discrete frequencies ($\hat{z}_{(f)}$, e.g. $\hat{z}_{M_2}$) for which amplitudes and phases can be calculated using the real and
imaginary parts. The approach we use in our work (e.g. HALS) was comprehensively tested showing that it leads to accurate
estimates of harmonics in the presence of noise levels not exceeding the measurement resolution (Schweizer et al., 2021). High
quality pressure transducers generally fulfil this criteria (Rau et al., 2019).

## 2.2 Earth tide influences on well water levels

### 2.2.1 Subsurface strain response to gravity changes

Rojstaczer and Agnew (1989) argued that for Earth tides horizontal areal strain is a sufficient approximation for depths of up to
thousands of kilometres. This approximation is sufficient for application to groundwater resources as they are generally much
shallower. The strain is often referred to as dilatation which is the total increase in volume of the material due to forcing by
the Earth tides (in this case the tidal pull). In porous media, assuming incompressible grains, this dilatation is manifesting as
an opening of the total pore space, decreasing the water pressure within the material (Agnew, 2010). In this paper the term
'dilatation' is used broadly for both the dilation and compression due to the cyclical forcing of the tides, coherent with its
previous literature use (Xue et al., 2016; Allègre et al., 2016). The distortions by dilatation can be estimated through the planar
strain concept known as tidal dilatation (Schulze et al., 2000; Fuentes-Arreazola et al., 2018). Tidal dilatation can be defined
as

$$e^t = \frac{V}{g} \cdot \frac{e^v - 3e^h}{R} \tag{1}$$

where $e^t$ is the tidal dilatation strain (-), in this instance at the $M_2$ frequency, $g$ is acceleration due to gravity ($\approx 9.81 m/s^2$),
$e^v$ is vertical displacement strain (-), $e^h$ is horizontal displacement (-) (Agnew, 2010), $R$ the average radius of the Earth ($m$)
adjusted for any significant elevation and $V$ is the tidal potential ($m^2 s^{-2}$) as defined in Table 1. The term ($e^v - 3e^h$) may
also be approximated by Love-Shida numbers where $e^v$ can be replaced by $_S^L h$ with an assumed value of 0.6032 and $e^h$
may be replaced with $_S^L l$ with an assumed value of 0.0839 (Agnew, 2010; Cutillo and Bredehoeft, 2011). Previous work has
demonstrated the use of calculated strain for analysing the groundwater response to Earth tide forces Roeloffs (1996); Xue et al.
(2016); Allègre et al. (2016); McMillan et al. (2019). As such, the terms $e^v$ and $e^h$ can be directly calculated from software
that generates theoretical Earth tides or tidal dilatation strains based on geo-location, for example *ETERNA* (Wenzel, 1996),
*TSoft* (Van Camp and Vauterin, 2005), or, as was done for this paper, using *PyGTide* (Rau, 2018) (based on ETERNA).

The first approach using ET to estimate specific storage, used the potential for water movement from the tides to the corre-
sponding water movement in a monitoring well in a confined aquifer for undrained conditions. Here, Bredehoeft (1967) defined
specific storage ($S_s$) for a medium that is presumed to be incompressible as

$$S_s = -\left[\left(\frac{1-2v}{1-v}\right)\left(\frac{2_S^L h - 6_S^L l}{R \cdot g}\right)\right]\frac{\Delta A_{M_2}^{ETp}}{\Delta h}, \tag{2}$$

where $\Delta A_{M_2}^{ETp}$ is the change in the tidal potential to the corresponding change in hydraulic head $\Delta h$ and $v$ is typically an
assumed Poisson's ratio. Here, the tidal dilatation (Equation 1), has been incorporated into Equation 2. Equation 2 was then





used by Cutillo and Bredehoeft (2011) for is advantages over methods such as provided by Hsieh et al. (1987), as it does not require the separation of individual tidal components, or the knowledge of a well's dimensions. Progressive improvements in the precision and duration of gravity measurement methods have since allowed for more accurate decomposition and cataloguing of the various tidal components (Agnew, 2010). These established catalogues of precise frequencies provide the basis for component separation using harmonic filtering techniques. The full separation of ET and AT at one frequency allows their individual and combined use towards better in-situ hydrogeomechanical characterisation (Rau et al., 2020).

### 2.2.2 Well water level response to harmonically forced pore pressure

In this paper, we focus on the ET component at the frequency of 1.932274 ($cpd$) cycles per day (denoted by a subscript of its Darwin name $M_2$) and the combined ET and AT component at the frequency of 2 $cpd$ (denoted by a subscript of its Darwin name $S_2$), described in Table 1. While other frequency components can also be used (Hsieh et al., 1988; Merritt, 2004; Cutillo and Bredehoeft, 2011), $M_2$ and $S_2$ generally have the strongest tidal impact and their forces remains constant over time (Acworth et al., 2016; McMillan et al., 2019).

The relative amplitude response ($A_{M_2}^e$) of the groundwater, as measured in a borehole in relation to the tidal dilatation strain can be expressed as (Hsieh et al., 1987; Xue et al., 2016; Allègre et al., 2016)

$$A_{M_2}^e = \left| \frac{\hat{z}_{M_2}^{GW}}{\hat{z}_{M_2}^{ETe}} \right| = \frac{A_{M_2}^{GW}}{A_{M_2}^{ETe}}, \tag{3}$$

where $\hat{z}_{M_2}^{GW}$ and $\hat{z}_{M_2}^{ETe}$ are the complex frequency component of the groundwater pressure head and tidal dilatation strain, respectively; $A_{M_2}^{GW}$ is the amplitude of the groundwater pressure head fluctuation and $A_{M_2}^{ETe}$ is the amplitude of the tidal dilatation strain fluctuation, all at the frequency of the $M_2$ tidal component. Note that $A_{M_2}^e$ is also referred to as areal strain sensitivity (Hsieh et al., 1987).

It is important to note the difference presented in Equation 3 from Xue et al. (2016) with the original dimensionless amplitude response calculated by Hsieh et al. (1987) as

$$A_{M_2} = \left| \frac{\hat{z}_{M_2}^{GW}}{\hat{z}_{M_2}^{p}} \right| = A_{M_2}^e S_s, \tag{4}$$

where $\hat{z}_{M_2}^{p}$ is the complex aquifer pore pressure response (superscript $p$ reflects pore). In this equation, the denominator term has changed from the complex amplitude of the pressure fluctuation with the tidal dilatation, effectively incorporating Equation 2. This key difference allows for the addition of the term $S_s$ within the amplitude response equations due to the sensitivity of storage to the amplitude response for post- and pre-strain responses described in Sections 2.2.3 and 2.2.4. $A_{M_2}$ from Equation 4 is dimensionless, with values $0 \leq A_{M_2} \leq 1$.

The phase shift (or difference) is defined as the strain response observed as the complex groundwater pressure head (water level) fluctuation, minus the phase of the complex tidal dilation (tidal forcing) stress, defined as

$$\Delta\phi_{M_2} = \arg\left( \frac{\hat{z}_{M_2}^{GW}}{\hat{z}_{M_2}^{ETe}} \right) = \phi_{M_2}^{GW} - \phi_{M_2}^{ETe}, \tag{5}$$





where $\phi_{M_2}^{GW}$ is the phase angle expressed in the groundwater response and $\phi_{M_2}^{ETe}$ is the phase angle of the theoretical Earth tide component, in this case at the frequency of the $M_2$. A negative phase shift is expressed where the groundwater response lags behind the induced strain (water level response lags behind the pressure head disturbance (Hsieh et al., 1987)), whereas a positive phase shift indicates the groundwater response is leading the strain response.

   It should be noted that in this method development, a homogeneous, isotropic aquifer of infinite lateral extent is assumed
for all derivations (Hsieh et al., 1987). All derived hydro-geomechanical variables are treated as bulk properties (averaged over a distinct but unknown volume), representative of the EAT area of influence around the monitoring wells screened interval, including effects from geological heterogeneities and the well construction, such as the inclusion of a gravel pack. The exact nature and dimensions of the volume of influence (i.e. the volume of sub-surface around the well being 'sampled') is currently unresolved. It is commonly assumed that the ET amplitude response is negligibly influenced by fluid flow when confined
(Xue et al., 2016); instead, it is predominantly controlled by the storage response. This is used as a justification to modify the first hydraulic diffusivity term in the amplitude response equations to $1/Ss$ when including the Earth tide strain estimation (Equations 6 and 13), i.e. the tidal dilatation (Hsieh et al., 1987; Wang, 2000; Xue et al., 2016).

### 2.2.3   Post-strain water level response

   Positive and negative phase shifts are either leading (pre-strain) or lagging (post-strain), respectively, in relation to the strain
response expressed by the water level in a well to formation tidal forcing. Hsieh et al. (1987) provided an analytical solution for the confined groundwater flow equation with harmonic forcing to describe the relationship between aquifer pore pressure and well water level. Their model is formulated in terms of amplitude ratio and phase shift, thereby allowing for the solution of two properties, transmissivity and storativity from the amplitude and phase response. This model works by exploiting the lack of sensitivity to storage within the phase shift equation and iterates to fit for both transmissivity and storage (See Figure
3) (Rau et al., 2020). The post-strain (negative phase) model is defined by Hsieh et al. (1988) as

$$A_{M_2}^e = \frac{1}{S_s}(E^2 + F^2)^{-\frac{1}{2}} \tag{6}$$

and

$$\Delta\phi_{M_2} = -\tan^{-1}\left(\frac{F}{E}\right) \tag{7}$$

where

$$E = 1 - \frac{\omega r_c^2}{2T}[\Psi Ker(\alpha_w) + \psi Kei(\alpha_w)] \tag{8}$$

and

$$F = \frac{\omega r_c^2}{2T}[\psi Ker(\alpha_w) - \Psi Kei(\alpha_w)] \tag{9}$$

and

$$\Psi = \frac{-[Ker_1(\alpha_w) - Kei_1(\alpha_w)]}{2^{\frac{1}{2}}\alpha_w[Ker_1^2(\alpha_w) + Kei_1^2(\alpha_w)]} \tag{10}$$





and

$$\psi = \frac{-[Ker_1(\alpha_w) + Kei_1(\alpha_w)]}{2^{\frac{1}{2}}\alpha_w[Ker_1^2(\alpha_w) + Kei_1^2(\alpha_w)]} \tag{11}$$

where

$$\alpha_w = r_w\sqrt{\frac{\omega S}{T}} = r_w\sqrt{\frac{\omega}{D_h}}. \tag{12}$$

The storativity $S$ and transmissivity $T$ can be related to specific storage as $S = S_s b$ and hydraulic conductivity as $T = k^f b$,

respectively; $b$ is the aquifer thickness, here is related to the vertical screen length when the aquifer thickness is unknown; $r_w$ is the internal radius of the well screen (accounts for well storage); $r_c$ is the radius of the casing. $Ker$ and $Kei$ are Kelvin functions of zero order, and $Ker_1$ and $Kei_1$ are Kelvin functions of the first order.

### 2.2.4 Pre-strain water level response

The pre-strain water level model is based on the description of a periodic load on a half-space, as described by Wang (2000),

and is used for Earth tides where a vertical head gradient exist (Xue et al., 2016; Allègre et al., 2016). The Equations 13 and 14 were derived from the force equilibrium equations (refer to Wang (2000))

$$A_{M_2}^e = \frac{1}{S_s}\sqrt{1 - 2\exp\left(-\frac{z}{\delta}\right)\cos\left(\frac{z}{\delta}\right) + \exp\left(-2\frac{z}{\delta}\right)}, \tag{13}$$

and

$$\Delta\phi_{M_2} = \tan^{-1}\left[\frac{\exp\left(-\frac{z}{\delta}\right)\sin\left(\frac{z}{\delta}\right)}{1 - \exp\left(-\frac{z}{\delta}\right)\cos\left(\frac{z}{\delta}\right)}\right], \tag{14}$$

where $z$ is depth of the midpoint open screen interval, $\omega$ is the angular frequency of the tidal component ($M_2$),

$$\delta = \sqrt{\frac{2D_h}{\omega}}, \tag{15}$$

and $D_h$ is then the hydraulic diffusivity, defined as

$$D_h = \frac{T}{S} = \frac{k^f}{S_s} = \frac{k}{\mu S} = \frac{\rho_w g k^f}{\mu S_s^p} \tag{16}$$

where $T$ is subsurface transmissivity ($T = k^f b$), $k$ is permeability, $k^f$ is hydraulic conductivity, $\rho_w$ is the density of water

(0.9982 $kg/L$ at $20°C$) and $\mu$ is the dynamic viscosity of water ($8.90 \cdot 10^{-4}$ Pa s), $S$ is storativity and $S_s^p$ is specific storage ($1/Pa$). Equations 13 and 14 require iterative solving for $D_h$ and $S_s$.

Both Equation 13 and 14 were developed for harmonic loading (i.e. ocean or barometric loading) where strain is produced at the surface of the Earth's crust and propagated down (Wang and Davis, 1996). ET (tidal dilatation) on the other hand, manifests within the subsurface where the stress is depth independent. Close attention is therefore required for the effect of depth when

analysing combined ET and AT forcing effects (rather than just a loading), ensuring that the sensitivity to depth has adequately attenuated (e.g. deeper than 10 m), as shown in Figure 2.

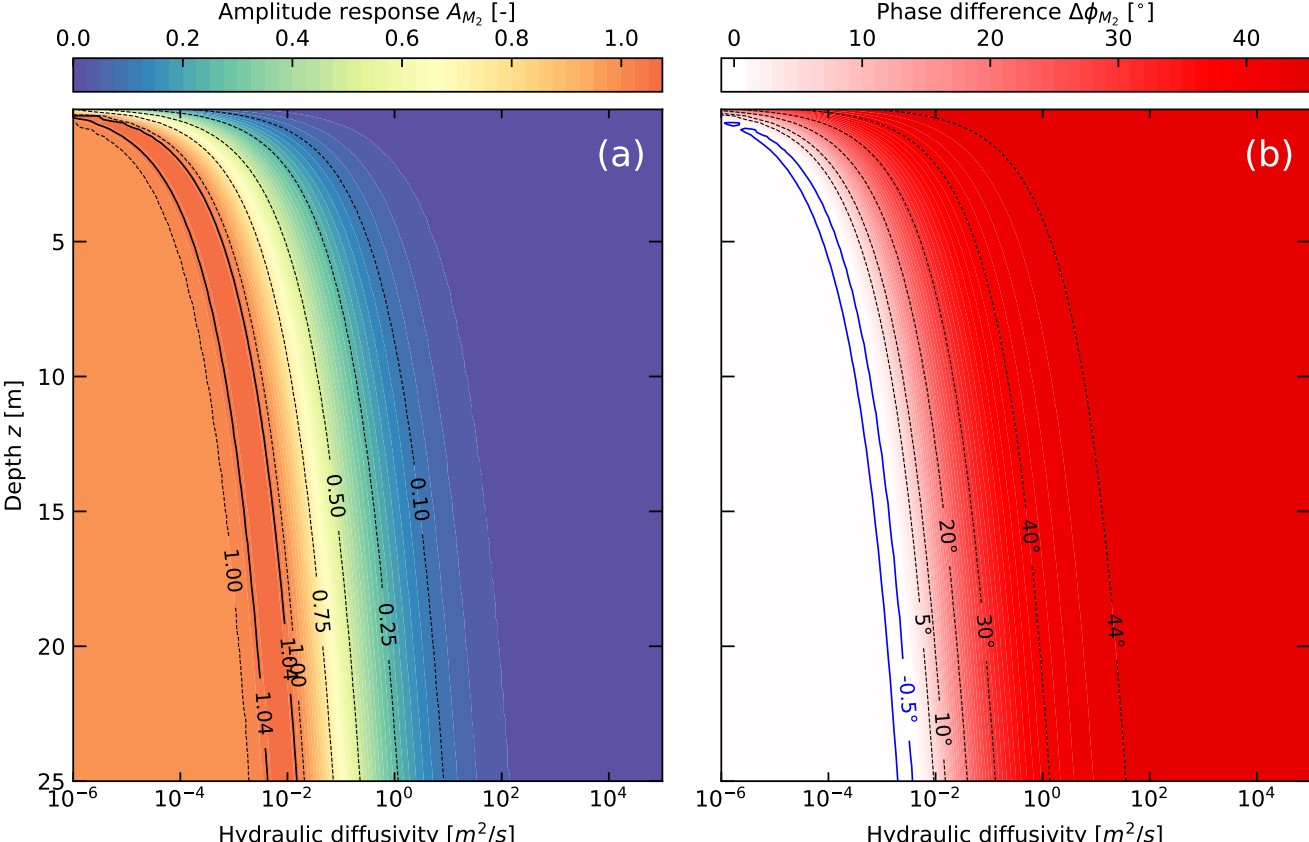

**Figure 2.** Periodic loading of a half space (applied to ET) as modeled by Equations 13 and 14. (a) Normalized relative amplitude response and hydraulic diffusivity as a function of depth (Equation 13). (b) Phase response and hydraulic diffusivity as a function of depth (Equation 14) (Wang, 2000).

### 2.2.5 Distinguishing between pre- and post-strain conditions

The sets of Equations 6 and 7 (Hsieh et al., 1987) describe lateral water movement between the well-bore and surrounding subsurface, whereas Equations 13 and 14 (Wang, 2000) explain the positive phase shift by allowing vertical flow within the groundwater system. Both sets of equations have been used to estimate hydraulic conductivity and specific storage. This is achieved by decomposing the hydraulic diffusivity using the assumptions outlined at the end of Section 2.2.2.

The phase shift determines which of these sets of analytical solutions are appropriate. For a phase between $0°$ to $-45°$ the post-strain response model is used, and for a phase between $-1°$ and $90°$ the pre-strain response model is applied (both are visualised in Figure 3). Note that the pre-strain model results in a slight negative phase shift for certain parameter ranges. Consequently, there is a range of ambiguity between phase shift values between $-1°$ to $0°$ in which both sets of solutions should be used, and the most physically plausible results should be selected (Xue et al., 2016). Note, the unit input as either





pressure or hydraulic head will also be carried through the equations resulting in a unit difference where $S_s^p$ is specific storage as pressure ($1/Pa$) whereas $S_s$ is specific storage as a reciprocal meter length ($1/m$), as demonstrated in equation 16.

**Figure 3.** Pressure head amplitude and phase response to the Earth tide $M_2$ component as a function of ranges in hydraulic conductivity and specific storage: (a) amplitude (Equation 6) and (b) phase (Equation 7) response for confined conditions (here; the radius of borehole and screen are 0.1 m and the screen length is 2 m). (c) amplitude (Equation 13) and (d) phase (Equation 14) response for semi-confined conditions where vertical flow may exist (depth of screen is 20 m).

## 2.3 Atmospheric tide influences on well water levels

Methods that quantify the barometric efficiency of subsurface systems are based on quantifying the groundwater response magnitude to atmospheric pressure changes (Clark, 1967; Rasmussen and Crawford, 1997; Barr et al., 2000; Gonthier, 2003) or atmospheric tides (Acworth et al., 2016). Turnadge et al. (2019) reviewed these methods and concluded that the method by Acworth et al. (2016) was the most robust and reliable. However, their approach was limited by the assumption of an





instantaneous and undamped response. Rau et al. (2020) developed a new method that completely disentangles the influences
of Earth and atmospheric tides at the same frequency, e.g. $S_2$. This further considers the damping of the subsurface-well system
that can be caused by low hydraulic conductivity materials. Their new approach is (Rau et al., 2020)

$$BE_{S_2} = \frac{1}{A_{M_2}} \cdot \left| \frac{\hat{z}_{S_2}^{GW.AT}}{\hat{z}_{S_2}^{AT}} \right|,$$  (17)

where

$$\hat{z}_{S_2}^{GW.AT} = \hat{z}_{S_2}^{GW} - \hat{z}_{S_2}^{GW.ET} = \hat{z}_{S_2}^{GW} - \frac{\hat{z}_{M_2}^{GW}}{\hat{z}_{M_2}^{ET}} \hat{z}_{S_2}^{ET}.$$  (18)

Here, $A_{M_2}$ corrects for the damping of the subsurface-well system, e.g. for low hydraulic conductivity, and can be inferred
from Earth tides as calculated earlier (Equation 4); $\hat{z}_{S_2}^{GW.AT}$ is the $S_2$ component of the groundwater response to atmospheric
tides, and $\hat{z}_{S_2}^{AT}$ is the $S_2$ frequency component (atmospheric tide) embedded in atmospheric pressure measurements. $BE$ forms
a stress balance, described as (Jacob, 1940)

$$BE + \gamma = 1,$$  (19)

where $\gamma$ is the loading efficiency.

### 2.4  Combining Earth and atmospheric tide responses

### 2.4.1  General relationships

Within the following derivations it is assumed that Earth tides only induce horizontal areal strain ($\epsilon_a = \epsilon_{11} + \epsilon_{22}$) whereas
atmospheric tides only induce vertical strain ($\epsilon_{33} = -p_{AT}$) (Rojstaczer and Agnew, 1989; Cutillo and Bredehoeft, 2011), all
of which are assumed to act instantaneously on the subsurface as is consistent with previous literature (Wang, 2000; Rau et al.,
2018). Under such conditions, van der Kamp and Gale (1983) has shown that the rigidity modulus (also known as the shear
modulus, $G$) can be estimated, with the previous outlined assumptions, from combined Earth and atmospheric influences as

$$G = A_{M_2}^e \frac{\rho g}{2\gamma} = A_{M_2}^e \frac{\rho g}{2(1 - BE)},$$  (20)

where, $A_{M_2}^e$ originates from Earth tides (Equation 3), whereas $BE$ or $\gamma$ is derived from atmospheric tides (Equation 19).
The disentanglement of Earth and atmospheric tides from the groundwater level response in a well, and the use of these
separate frequency components to quantify hydrogeomechanical properties allows further geomechanical derivations to be
made. Two methods are presented below which solve for the assumption of either incompressible (suitable for unconsolidated
material) or compressible grains (suitable for consolidated material). The choice between which method to use is established
by examining an estimated *Biot-Willis* coefficient defined as (Wang, 2000)

$$\alpha = 1 - \frac{K}{K_s} = 1 - \frac{\beta_s}{\beta}.$$  (21)





Where $K$ is the Bulk modulus, $K_s$ is the Bulk modulus of the solid grain; respectively, $\beta$ is the compressibility, and $\beta_s$ the compressibility of the solid grain. For unconsolidated conditions, where the Bulk modulus is much smaller than the Bulk modulus of the grains ($K \ll K_s$) it is possible to assume that the grains do not contribute to the overall compressibility, thus the grains are incompressible. The *Biot-Willis* coefficient $\alpha \to 1$ shows that the contribution of the grains to the compressibility

of the bulk material is insignificant (Rau et al., 2018). By contrast, in consolidated cases $K$ becomes larger, leading to a coefficient that deviates appreciably from one ($\alpha < 1$). In such cases, the grain compressibility is a significant proportion of the total material compressibility and must be accounted for.

### 2.4.2 Incompressible grains

For incompressible grains ($\alpha = 1$) the uniaxial loading efficiency is related to the uniaxial bulk properties as (van der Kamp
and Gale, 1983)

$$\gamma = \frac{\beta_v^u}{\theta \beta_f + \beta_v^u}, \tag{22}$$

where $\beta_f$ is the compressibility of the fluid ($4.59 \times 10^{-10} \ Pa^{-1}$ at $20°C$), $\beta_v^u$ is the vertical undrained bulk compressibility and $\theta$ is the total porosity of the formation. The uniaxial specific storage (assuming incompressible grains) is defined by Jacob (1940) as

$$S_s = \rho g(\beta_v^u + \theta \beta_f). \tag{23}$$

This equation was used by Acworth et al. (2016), with an $S_s$ estimate from Equation 24, to constrain Equation 23 allowing $\beta_v$ to be resolved.

$$S_s = \rho_w g \beta_f \frac{\theta}{BE} = 4.5 \times 10^{-6} \frac{\theta}{BE}. \tag{24}$$

However, this requires a prior estimate of the porosity $\theta$ which is often difficult to determine due to the lack of available
field measurements. Note also that the above equations assume that barometric loading is uniaxial, and as such use vertical compressibility ($\beta_v$) rather than that of the volumetric (bulk) compressibility ($\beta$). Here, were instead propose using the $S_s$ derived from the pre- or post-strain response to ET (Section 2.2) to instead constrain Equation 24 to estimate the subsurface porosity by rearranging Equation 24 (similar to Jacob (1940)) as

$$\theta = \frac{S_s BE}{\rho g \beta_f} = \frac{S_s}{\rho g \beta_f}(1 - \gamma). \tag{25}$$

To achieve a similar outcome as Acworth et al. (2016) this porosity, in addition to the calculated $S_s$, can now be used in Equation 26, rearranged from Acworth et al. (2016), to provide a uniaxial (vertical) bulk compressibility (inverse vertical undrained bulk modulus ($K_v^u$)) of the subsurface defined as (Acworth et al., 2016)

$$\beta_v^u = \frac{\gamma \theta \beta_f}{1 - \theta} = \frac{1}{K_v^u}. \tag{26}$$





This approach is similar to the one used by Cutillo and Bredehoeft (2011) but uses the objective $BE$ method developed by Rau et al. (2020) instead of the subjective correlation by Gonthier (2003). Within this subsection it has been shown that it is possible to derive an estimate of porosity from a loading strain if the specific storage is known. This assumes incompressible grains and is therefore suitable for unconsolidated material (Rau et al., 2019).

The assumption of incompressible grains allows for the removal of the grain compressibility and provide a simplification of the poroelastic space. This step, combined with the new derivation of the shear modulus enables a linear analytical solution of the remaining elastic variables in unconsolidated material ($\alpha \approx 1$). The first step can be taken by deriving the undrained bulk modulus ($K_u$) with the $K_v^u$ from Acworth et al. (2016) as (Wang, 2000)

$$K^u = K_v^u - \frac{4}{3}G, \tag{27}$$

which allows for the solving of Skempton coefficient defined as (Rau et al., 2018)

$$B = \gamma \frac{K_v^u}{K^u} = \gamma \frac{\beta^u}{\beta_v^u}. \tag{28}$$

Determination of the Skempton coefficient along with the loading efficiency unlocks the undrained Poisson's ratio using (Wang, 2000)

$$\nu^u = \frac{3\gamma - B}{3\gamma + B} \tag{29}$$

and drained Poisson's ratio as (Wang, 2000)

$$\nu = \frac{3\nu^u - B(1 + \nu^u)}{3 - 2B(1 + \nu^u)}. \tag{30}$$

Determination of the drained Poisson ratio further unlocks all remaining poroelastic properties such as Young's Modulus ($E$), defined as (Wang, 2000)

$$E = \frac{9KG}{3K + G}. \tag{31}$$

Equations 22-31 define the complete parameter space for unconsolidated materials.

### 2.4.3 Compressible grains

To determine the poroelastic properties of consolidated materials, the grain compressibility must be considered ($\alpha < 1$). Further, the following two assumptions must be made: (1) Although pore fluids technically respond to cubic strains, the areal strain can be used to estimate the subsurface strain from ET; (2) The system is homogeneous and laterally extensive, thus ignoring topographic effects and considering the barometric loading to be uniform. The equations that define the remaining elastic properties for such conditions are (Beavan et al., 1991)

$$B = \frac{3\gamma(1 - \nu)}{2\gamma\alpha(1 - 2\nu) + (1 + \nu)}, \tag{32}$$





and

$$\theta = \left(\frac{1}{B} - 1\right)\left(\frac{1}{K} - \frac{1}{K_s}\right)\left(\frac{1}{k^f} - \frac{1}{K_s}\right)^{-1}, \tag{33}$$

and

$$\alpha = 1 - \frac{K}{K_s} = 1 - \frac{2G(1+\nu)}{3K_s(1-2\nu)} \tag{34}$$

and

$$S_s = \frac{\rho g}{\gamma(1-\nu)}\left(\frac{1-2\nu}{2G} - \frac{1+\nu}{3K_s}\right). \tag{35}$$

Equations 32-35 form a non-linear system which must be solved by iteration.

If the petrology of the lithology is known, appropriate literature compressibility values of the dominant grain mineralogy ($K_s$) could be used. Quartz is the most common naturally occurring mineral and is also one of the least compressible (it is also
applicable for most of our case sites), and it will therefore be used to define the upper bounds of $K_s$ here. Richardson et al. (2002) summarised literature values of poly-crystalline quartz for $K_s$ to range between 36-40 GPa, and reported $K_s$ values for the quartz Ottawa Sandstone to be in a range of 30-50 GPa. The average of these ranges has been summarised as 42 GPa (Rau et al., 2018) and will be used in this work.

With the established inputs of $\gamma(BE)$, $A_{M_2}$, $G$ (Equation 20), $S_s$ and an estimate of $K_s$, it is possible to simultaneously
solve Equations 32-35 for Skempton's coefficient ($B$), porosity ($\theta$), *Biot-Willis* coefficient ($\alpha$) and specific storage ($S_s$) (Beavan et al., 1991). This allows a complete calculation of all remaining poroelastic properties whose inter-dependency is summarised in Table 2.

| | $K$ | $K_v$ | $\nu$ | $E$ | $G$ |
|---|---|---|---|---|---|
| | Bulk Modulus | Uniaxial Drained Bulk Modulus | Poisson's Ratio | Young's Modulus | Shear Modulus |
| $K,G$ | - | $K + \frac{4G}{3}$ | $\frac{3K-2G}{2(3K+G)}$ | $\frac{9KG}{3K+G}$ | - |
| $G,E$ | $\frac{EG}{3(3G-E)}$ | $G\frac{4G-E}{3G-E}$ | $\frac{E}{2G}-1$ | - | - |
| $E,K$ | - | $3K\frac{3K+E}{9K-E}$ | $\frac{3K-E}{6K}$ | - | $\frac{3KE}{9K-E}$ |
| $G,\nu$ | $G\frac{2(1+\nu)}{3(1-2\nu)}$ | $G\frac{2-2\nu}{1-2\nu}$ | - | $2G(1+\nu)$ | - |
| $K,\nu$ | - | $3K\frac{1-\nu}{1+\nu}$ | - | $3K(1-2\nu)$ | $3K\frac{1-2\nu}{2+2\nu}$ |
| $E,\nu$ | $\frac{E}{3(1-2\nu)}$ | $\frac{E(1-\nu)}{(1+\nu)(1-2\nu)}$ | - | - | $\frac{E}{2+2\nu}$ |

**Table 2.** Elastic constant relationships for isotropic stress and undrained conditions (Birch, 1996; Wang, 2000; Sheriff, 2002). Note that Young's modulus may also be used to provide the uniaxial compression strength (UCS) using the linear relationship established by Colwell and Frith (2006).





## 3 Method application under different hydrogeological settings

### 3.1 Field sites, geological context and monitoring

To demonstrate the new method, groundwater and barometric pressure records from four sites and five monitoring bores were used. These sites were selected based on three main criteria: (1) Data availability and quality; (2) a strong groundwater response to Earth tides ($M_2$); (3) providing a variety of hydrogeological settings with existing studies for parameter comparisons. The Cattle Lane site has unconsolidated materials and was processed using the approach for unconsolidated systems with the assumption of incompressible grains. All other sites were evaluated by assuming compressible grains. Specific bore geometries

and measurements used in the analysis of these sites such as depths and bore construction are summarised in Table 3.

#### 3.1.1 Cattle Lane (NSW, Australia)

Cattle Lane is located on the Liverpool Plains, NSW, eastern Australia. Erosion of the basaltic Liverpool Ranges to the south produced a succession of unconsolidated silts, clays, sands, gravel and minor carbonate nodules within the Liverpool Plains. A

thick sequence of clay bound sediments overlie a gravel aquifer at 40 m. This aquifer has previously be shown to respond to loading by rainfall events (Timms and Acworth, 2005). The lithology of the 1 m screened interval was described by Acworth et al. (2015) as major basalt fragments mixed with coarse sand, shell and carbonate nodules. The site has previously been cored to 31.5 $m$ depth, lithologically logged and geophysical surveyed, confirming that it is horizontally extensive (Acworth et al., 2015). Cross-hole seismics were also conducted by Rau et al. (2018) to the depth of 40 $m$ (screened interval of bore BH30061

is 55 m depth, see Table 3), providing depth profiles of seismically inferred elastic variables that were used to constrain the pore pressure response to atmospheric tides analysis.

Further studies at this site include Acworth et al. (2016) and Acworth et al. (2017), which were precursors to Rau et al. (2018) in the investigation of pore pressure response to atmospheric tides, and Timms et al. (2018) on a core scale analysis of the site's laterally extensive and thick aquitard. Due to the sufficient $M_2$ response, time-series data of groundwater pressure heads

measured between the 21/01/2016 and 14/04/2018 were used from the bore BH30061 which is located at latitude $-31.518340°$, longitude $150.468332°$ and an elevation of 313 MASL (WGS84). The groundwater pressure heads were collected using vented In-Situ Troll 700H series loggers at hourly intervals and sub-millimetre precision. Atmospheric pressure was measured by an In-Situ Baro Troll absolute gauge transducer.

#### 3.1.2 Thirlmere Lake (NSW, Australia)

Thirlmere Lakes is located in the south-west of the Sydney Basin, NSW. Both bores are located in the quartz arenite Hawkesbury sandstone, which is about 100 m thick at the site. This sandstone is deposited by a braided river with the heterogeneous deposits showing overlapping and self incised fining up sequences, with over-bank deposited fines at paleo-channel boundaries (Miall and Jones, 2003). There is evidence that the upper portion of bore Thirlmere 2 passes through a geological fault damage





| Location | Borehole | Inputs | | | | | | | | Description | |
| | | $A^e_{M_2}$ | $A_{M_2}$ | $\Delta\phi_{M_2}$ | $r_c$ | $r_w$ | $b$ | $z$ | $K_s$ | Lithology | Strain |
|---|---|---|---|---|---|---|---|---|---|---|---|
| Cattle Lane | BH30061 | $4.59 \cdot 10^4$ | 0.765 | 24.89 | 0.125 | 0.12 | 1 | 55 | - | Sand, Gravel, Clay | Pre |
| Thirlmere | GW075409.1.2 | $3.24 \cdot 10^5$ | 1.039 | 10.05 | 0.156 | 0.14 | 12 | 78 | 42 | Quartz arenite sandstone | Pre |
| Thirlmere | Thirlmere 2 | $3.52 \cdot 10^5$ | 0.988 | -4.74 | 0.114 | 0.108 | 4 | 72 | 42 | Quartz arenite sandstone | Post |
| Dodowa | BH11 | $1.31 \cdot 10^6$ | 0.948 | -13.8 | 0.058 | 0.048 | 2 | 45 | 42 | Gneiss | Post |
| Death Valley | BLM-1 | $1.49 \cdot 10^6$ | 0.998 | -1.07 | 0.127 | 0.127 | 106 | 830 | 42 | Carbonate | Post |

**Table 3.** Inputs parameters for case sites where; $A_{M_2}$ is the dimensionless amplitude response, $A^e$ is the amplitude response, $\phi_{M_2}$ is the phase shift of the $M_2$ component, $r_c$ is the outer diameter (m) of the bore casing, $r_w$ the internal diameter (m) of the bore casing, $b$ is the Aquifer thickness (m) or open interval of the screen, $z$ is the depth (m) to the centre of the screen or open interval, and $K_s$ is the assumed grain bulk modulus. Italicised values were not used in the applied pre- or post-strain models, but are provided for context.



zone, with drilling fluid losses recorded above the screened interval due to fractures (Impax, 2019). Other studies in the same
lithology include Ross (2014), which investigated the potential for a bore field development within the Hawkesbury Sandstone,
however, no publicly available studies exist for this lithology at the case site.

The time span and collection of the time-series data for the two bores differ. The time-series for GW075409.1.2 covers
the time period from 03/07/2018 to 14/12/2018 and was downloaded from the *WaterNSW* real-time data portal with 15 min
intervals. The bore is located at latitude $-34.230666^o$, longitude $150.543996^o$, elevation 314 MASL (Russell, 2012). For this
bore a coinciding barometric time-series data was obtained from a weather station approx. 500 m away. The bore Thirlmere
2 is located about 2 km from GW075409.1.2 and is located at latitude $-34.220836^o$, longitude $150.536467^o$, elevation 323
MASL. The time-series data for Thirlmere 2 was collected for this study using a vented In-Situ Troll 700H series pressure
transducer every 5 min between the 32/07/2019 and 29/10/2019. The coinciding barometric time-series was collected using a
Solinst Baro-logger installed in the airspace of the borehole.

### 3.1.3 Dodowa (Ghana)


Dodowa is located in the Shai Osudoku District in the southeastern part of the Greater Accra Region, Ghana. The local geology
consists of the Togo Structural and Dahomeyan Structural units. The Togo being composed of a series of metamorphic and
folded quartzites, phyllites and schists, and the Dahomeyan composed of altered belts of acid and basic gneisses. BH11 used
within this paper is located in a Dahomeyan gneiss (Attoh et al., 1997). All units within the region appear highly weathered,
resulting in an 5 m unconsolidated regolith, confining the underlying fractured igneous and metamorphic units.

BH11 was installed and previously studied by Foppen et al. (2020), including atmospheric tide analysis. The time-series for
the water levels of BH11 was collected at 20 min intervals between the 18/10/16 and 07/06/2017 using Mini-Diver DI501;
Schlumberger pressure transducers, with atmospheric pressure being recorded with a Mini-Diver DI500; Schlumberger baro-
metric diver, located above ground at the site at an approximate latitude $5.881675$, longitude $-0.097244$, elevation 88 MASL
(Foppen et al., 2020).

### 3.1.4 Death Valley (California, USA)

The Death Valley site is located in the western part of the USA on the border of Nevada and California at a position of; latitude
36.408130, longitude -116.471360 (WGS84), elevation 688 MASL. Bore BLM-1 is located in Paleozoic carbonate rock and
was left as an open well. The same time-series record was also used in Rau et al. (2020) and it is the same bore for which data
was analysed in Cutillo and Bredehoeft (2011). Data was recorded at 15 min intervals using an In Situ Troll with a vented cable
and an In Situ Barotroll. The time-series spans between the 25/06/2009 and 16/12/2009.

### 3.2 Method application

Groundwater pressure head and barometric pressure time-series were recorded at sub-hourly intervals at all sites (e.g. two sites
shown in Figure 4) for at least three months which is longer than the $\sim$28 days being suggested as the minimum requirement





(Roeloffs, 1996). The theoretical Earth tide potential for the same duration and sampling frequency of each site was calculated

using *PyGTide* (Rau, 2018). This required knowledge of the geo-position of the borehole (latitude, longitude and elevation in

WGS84). Additional information required for the analysis, such as casing and screen radius's, screen depth and length, were

also noted for each bore and were presented in Table 3. All time-series were detrended by a moving 3 day average using the

*SciPy* detrend function, and the main tidal components were extracted using *HALS* (Section 2.1).

**Figure 4.** Time-series of barometric pressures (m), theoretical Earth tide nano-strain (nstr) from *PyGTide* for the corresponding groundwater

levels from bores GW075409.1.2 and Thirlmere 2 from Thirlmere Lakes, NSW, Australia.

The following offers a step-by-step summary of the method:



1. Calculate the theoretical Earth tide potential time-series for the location and same time duration and interval of collected groundwater and barometric pressure data with reference to Coordinated Universal Time (UTC).

2. Extract the dominant tidal components using HALS (Schweizer et al., 2021) for barometric and groundwater pressure heads as well as Earth tide potential. Only the $M_2$ and $S_2$ are used in this work.

3. For the $M_2$ component, convert the tidal potential to dilatation strain and calculate the amplitude responses.

4. Compute the phase shift between the groundwater and Earth tides. A negative phase shift points to a post-strain groundwater response and the Equations 6 and 7 from Section 2.2.3 should be used. A positive phase shift indicate a pre-strain response and Equations 13 and 14 from Section 2.2.4 should be used. Evaluate hydraulic conductivity ($k^f$) and specific storage ($S_s$) using either the post or pre-strain models described in Sections 2.2.3 or 2.2.4 depending on negative or

positive $M_2$ phase shift between ET and GW respectively, as shown in Figure 5.

5. Calculate the barometric efficiency (Equation 17) using the normalised amplitude response (Equation 4), and the shear modulus (Equation 20) using the amplitude response to tidal dilation strain (Equation 3).

6. Distinguish between unconsolidated and consolidated systems:

   a Unconsolidated: This assumes incompressible grains ($K_s \rightarrow \infty$ and $\alpha = 1$). $BE$ is then combined with the specific

storage output from either Section 2.2.3 or 2.2.4 (depending on whether the phase is positive or negative) to solve for porosity using Equation 25 assuming incompressible grains.

   b Consolidated: This assumes compressible grains ($K_s < \infty$ and $\alpha < 1$). For consolidated conditions, the $BE$ is converted to a loading efficiency by using Equation 19, a shear modulus derived using Equation 20, and combined with the specific storage, dilatation strain, and an assumed solid grain compressibility to simultaneously solve

Equations 32 to 35.

7. All remaining poroelastic properties whose relationships are shown in Table 2 may then be derived, e.g. Young's modulus ($E$) using Equation 31.

In this paper, all of the methodology and equations were implemented in the *Python* programming language, and joint iterative solving was completed with *SciPy*'s *curve_fit* function which minimises the least-squares error. The following realistic

parameter bounds were considered for fitting: $0 \leq B \leq 1$, $-1 \leq \nu \leq 0.5$, $0.005 \leq G \leq 40$ GPa, $0 \leq \theta \leq 0.5$. We note that (1) the parameter units were scaled to avoid bias towards parameters with large values, (2) the solver was set to 64-bit machine precision (epsilon $1.11 \cdot 10^{-16}$), (3) none of the parameters quantified in this work exceeded any of the prescribed fitting bounds.



**Figure 5.** Phase and amplitude responses from the processing of bore Thirlmere 2; a) and b) plot (black dot) the amplitude ratio and phase shift relationships between the subsurface pore pressure and well water level for the post-strain Earth tide model (Section 2.2.3), c) and d) are polar plots showing the amplitude and phases of the complex inference of the well response to Earth tides from the response at $M_2$, and the disentanglement of the well response at atmospheric tide $S_2$, respectively.





### 3.3 Hydro-geomechanical properties

The hydro-geomechanical properties for the field sites from the application of the method outlined in Section 3.2 are presented
in Table 4. The boreholes BH30061 and GW075409.1.2 from Cattle Lane and Thirlmere Lakes produced positive $M_2$ phase
shifts (Table 3), with specific storage and hydraulic conductivity therefore being derived from the Pre-strain model (Section
2.2.4). All other bores had negative phase shifts and were processed using the Post-strain model (Section 2.2.3) from Hsieh
et al. (1987). BH30061 was the only data set processed using the proposed unconsolidated analytical model. If applying the
assumed grain compressibility of quartz ($K = 42$ GPa) for BH30061, a *Biot-Willis* coefficient of 0.99 is obtained and hence
justifies the assumption of incompressible grains ($\alpha \approx 1$) (Section 2.4.2). Both the quartz sandstone bores returned *Biot-Willis*
coefficients of 0.96, and the gneiss bore 0.84, as such, these bores required a value for the grain compressibility (Section 2.4.3).

### 3.3.1 Cattle Lane (NSW, Australia)

The specific storage, hydraulic conductivity, porosity, shear modulus, and undrained Poisson's ratio from Cattle Lane are
consistent with literature values for the sediment type (Bowles, 1996), and comply with previous estimates from higher in
the stratigraphy at the same site obtained by cross-hole seismics (Acworth et al., 2015, 2016; Rau et al., 2018). The Young's
modulus of 0.4 GPa deviates from the expected material range reported in the literature for an unconsolidated clay, sand and
gravel mixture of between 0.025 and 0.2 GPa, although is reasonable when considering consolidation at 55 m depth and an
in-situ derivation rather than laboratory tests (Bouzalakos et al., 2016). The Poisson's Ratio of $-0.29$ is the only parameter that
deviates significantly from the expected range of 0.2 to 0.5. This will be discussed later.

### 3.3.2 Thirlmere Lakes (NSW, Australia)

Estimates of hydro-geomechanical parameters ($S_s$ of $3.2 \cdot 10^{-6}$ and $2.8 \cdot 10^{-6}$ (1/m); $k^f$ of $4.8 \cdot 10^{-7}$ and $1.9 \cdot 10^{-5}$ (m/s))
for the two sandstone bores are considered realistic for a quartz sandstone in this area. The higher $k^f$ for Thirlmere 2 is
believed to be indicative of enhanced hydraulic conductivity due to fractures. For this sandstone formation, SCA (2005, 2006)
has previously reported $S_s$ values of $2.49 \cdot 10^{-6}$ to $2.41 \cdot 10^{-4}$ (1/m) and $k^f$ of $1.15 \cdot 10^{-6}$ to $3.36 \cdot 10^{-6}$ (m/s) within this
formation, including fracture networks (Ross, 2014). Geomechanical estimates of the shear modulus of 2.6 GPa marginally
exceeds the expected range of $1 - 2$ GPa (Bertuzzi, 2014; Zhang et al., 2016; Zhang and Lu, 2018). Conversely, the bulk
modulus and Young's modulus both fall within the expected ranges of 2.6 to 5.3 and 3 to 8 GPa, respectively. The estimated
Poisson's ratios of $-0.64$ and $-0.03$ are low compared to values between 0.2 and 0.3 typically measured in the laboratory
(McMillan et al., 2019).

### 3.3.3 Dodowa (Ghana)

The hydrogeomechanical estimates of hydraulic conductivity of $2.6 \cdot 10^{-6}$ (m/s) and specific storage of $7.2 \cdot 10^{-7}$ (1/m) are
comparable with the values for the Togo Structural Unit from Foppen et al. (2020) derived from pumping and slug tests, which





| Location | Borehole | Results | | | | | | | | | | | | | |
|---|---|---|---|---|---|---|---|---|---|---|---|---|---|---|---|
| | | $<K>$ | $S_s$ | $BE$ | $K_s$ | $\theta$ | $K_v^u$ | $G$ | $K^u$ | $B$ | $\nu^u$ | $\nu$ | $K$ | $E$ | $\alpha$ |
| Cattle Lane | BH30061 | $5.4 \cdot 10^{-6}$ | $1.7 \cdot 10^{-5}$ | 0.11 | $\infty$ | 0.39 | 3.8 | 0.3 | 3.5 | 1.0 | 0.46 | -0.29 | 0.1 | 0.4 | 1 |
| Thirlmere | GW075409.1.2 | $4.8 \cdot 10^{-7}$ | $3.2 \cdot 10^{-6}$ | 0.38 | $\infty$ | 0.27 | 9.7 | 2.6 | 6.3 | 1.0 | 0.32 | -0.64 | 0.3 | 1.9 | 1 |
| Thirlmere | Thirlmere 2 | $1.9 \cdot 10^{-5}$ | $2.8 \cdot 10^{-6}$ | 0.35 | 42 | 0.19 | 16.6 | 2.6 | 13.1 | 0.9 | 0.38 | -0.03 | 1.6 | 5.1 | 0.96 |
| Dodowa | BH11 | $2.6 \cdot 10^{-6}$ | $7.2 \cdot 10^{-7}$ | 0.70 | 42 | 0.08 | 58.8 | 21.6 | 30.0 | 0.7 | 0.10 | -0.21 | 8.1 | 34.2 | 0.81 |
| Death Valley | BLM-1 | $4.3 \cdot 10^{-6}$ | $6.7 \cdot 10^{-7}$ | 0.62 | 42 | 0.06 | 64.2 | 19.1 | 38.7 | 0.8 | 0.16 | -0.23 | 6.7 | 29.4 | 0.84 |

**Table 4.** Results from case sites where; $<K>$ is hydraulic conductivity (m/s), $S_s$ is the specific storage (1/m), $BE$ is the barometric efficiency, $K_s$ is the solid grain bulk modulus (assumed to be 42 GPa for consolidated systems, else $\infty$), $\theta$ is porosity (-), $K_v^u$ is the vertical undrained bulk modulus, $G$ is the shear modulus (GPa), $K^u$ is the undrained bulk modulus, $B$ is *Skempton's* coefficient (-), $\nu^u$ is the undrained Poisson's Ratio (-), $\nu$ is Poisson's Ratio (-), $K$ is Bulk Modulus (GPa), $E$ is Young's modulus (GPa) and $\alpha$ is the *Biot-Willis* coefficient (-).





indicated ranges between $10^{-5}$ to $10^{-6}$ $(m/s)$, and $2.3 \cdot 10^{-7}$ to $7.7 \cdot 10^{-8}$ $(1/m)$, respectively. The estimated porosity of 0.08 for BH11 slightly exceeds the range of 0.005 to 0.05 in Foppen et al. (2020). Comparison of elastic modulus is problematic for schists, as values are dependent on the original protolith and may vary significantly, and because schistose rock masses are known for high values of anisotropy (Hoek and Diederichs, 2006). For example, Young's modulus for a schist, as in the screened interval of BH11, can vary significantly between 21 to 117 GPa depending on mineralogy and foliation orientation

(Condon et al., 2020). Our estimated value of 34.2 GPa falls within this range. However, detailed mineralogy does not exist for this bore to allow a closer comparison with literature values.

### 3.3.4   Death Valley (California, USA)

The estimated hydraulic conductivity of $4.3 \cdot 10^{-6}$ $(m/s)$, is in agreement with the Earth tide analysis derived value of $1.3 \cdot 10^{-6}$ $(m/s)$ by Cutillo and Bredehoeft (2011). In contrast, the estimated specific storage value of $6.7 \cdot 10^{-7}$ $(1/m)$ is an order of

magnitude smaller than the value of $7.3 \cdot 10^{-6}$ $(1/m)$ from Cutillo and Bredehoeft (2011). However, the specific storage and hydraulic conductivity values are both consistent with the values published by Rau et al. (2020) for the same data-set, using a method based on ET. The determined porosity (0.06) also aligns with the lower end of the range proposed by Cutillo and Bredehoeft (2011), and it is reasonable to assume the calculated Young's and shear modulus of 28.28 and 24.10 GPa are similarly plausible coinciding with literature lithological values (Parent et al., 2015). We note that the derived Poisson's ratio

of -0.23 differs significantly from the value of 0.25 which was merely assumed in Cutillo and Bredehoeft (2011).

## 4   Discussion

### 4.1   Influences on the parameter estimation

While quantifying hydro-geomechanical properties it is beneficial to consider factors that influence the accuracy of parameter estimation. We note that absolute errors in the measurement of atmospheric or groundwater pressure are irrelevant for the

presented methodology. This is because all parameters are based on the amplitude and phase of tidal components embedded within the measurements, i.e., the relative change characteristic of the harmonics. Here, the resolution of the measurement device directly determines detection and quantification of the responses to tidal forces. Schweizer et al. (2021) demonstrated that extraction of harmonics using HALS is accurate if the resolution of the pressure transducer is larger than twice the amplitude of the tidal component under consideration. All instruments deployed in the field examples of this work comfortably fulfilled

this criterion. Schweizer et al. (2021) further noted that HALS outperforms the discrete Fourier transform, but also that an objective error estimation for HALS is difficult and depends on the nature of the residuals (difference between measurement and model). We consider that the accuracy of HALS is at least as good as that resulting from fitting a conceptual model to measurements obtained during standard hydraulic testing such as aquifer testing.

     Previous works have illustrated that quantifying $BE$ by disentangling the groundwater response to EAT based on theoretical

Earth tides does not lead to additional uncertainty in parameter estimation since it evaluates the relative responses between





ET and GW (Acworth et al., 2016). However, this observation is valid only in a subsurface where the hydraulic conductivity is $< K > \gtrsim 1 \cdot 10^{-5}$ m/s (Rau et al., 2020). The borehole water level response in lower $< K >$ environments is damped and shifted compared to the pore pressure response outside the bore. A correction requires knowledge of both $< K >$ and $S_s$ which can be quantified using calculated ET strains. While this has been done before (Hsieh et al., 1987; Xue et al., 2016; Allègre et al., 2016; Rau et al., 2020), there is no literature investigating its associated uncertainties.

### 4.2 Harmonic disentanglement allows estimation of the poroelastic parameter space

In this work, we make use of recent advances that allow quantitative disentanglement of the groundwater response to both Earth and atmospheric tidal forces. Since each mechanism acts differently on the subsurface, the disentangled responses can be merged through theoretical relationships. Unlike previous research, this allows the solving of the complete poroelastic space for unconsolidated systems entirely based on time-series of measured groundwater pressure heads, atmospheric pressure, and calculated Earth tides. For consolidated systems, the complete poroelastic space can also be solved through a system of nonlinear equations by assuming the grain compressibility. This approach has previously been used in Rau et al. (2018).

A general agreement is held throughout the literature that a negative phase shift is representative of an observed time lag caused by the slow flow of fluid from the formation into the bore, in response to the tidal strain (Bower, 1983; Hsieh et al., 1987; Kümpel, 1997; Schulze et al., 2000). Conversely, no such agreement is held for positive phase shifts. Although Section 2.2.4 is based on the assumption that positive phase shifts relate to vertical flow to the water table, i.e. semi-confined conditions (Roeloffs et al., 1989), other explanations for positive phase shifts exist within the literature. These include the influence of fracture transmissivity and length, ocean loading, heterogeneous material properties and topographic effects (Roeloffs, 1996; Burbey, 2010). Here, positive phases from either vertical flow or fracture flow describe a process in which pressure is able to be propagated rapidly, either to the water table or along a highly transmissive fracture (Bower, 1983). Other mechanisms for phase shifts have also been explored in the broader literature, such as Hanson and Owen (1982), who related fracture orientation (strike and dip) to either positive or negative phase shifts.

In this study, positive and negative phases shifts were recorded at the various field sites. A comprehensive understanding of negative and particularly positive phase shifts is still lacking within the literature. Shi and Wang (2016) observed that negative phase shifts were indicative of predominantly horizontal groundwater flow in a completely undrained system, while a positive phase shift was indicative of a vertical hydraulic gradient in a semi-confined or unconfined system. The method by Hsieh et al. (1987) outlined above as the post-stain model (negative phase shift), which was used by Shi and Wang (2016), is based on the assumption of radial horizontal flow into a well. If a positive rather than negative phase shift is used as an input into the system of equations provided by Hsieh et al. (1987), the results will not be sensible. As such, a positive phase shift model is required. For this project the method provided by Wang (2000), and adapted by Xue et al. (2016) and Allègre et al. (2016), was implemented to account for vertical flow. This method, as described in Section 2.2.4, was developed for a subsurface forcing by harmonic surface loading. Earth tides do not act by surface loading but rather the mechanism is tidal dilatation, where gravitational forces act on mass across the vertical profile. Although the method based on positive phase shifts has been successfully applied within the literature, the validity of this model has not yet been proven and further research is necessary.



Previous methods that utilised EAT for subsurface characterisation have always required the assumption of an elastic modulus, typically Poisson's ratio, to resolve additional geomechanical parameters of the subsurface. The outlined method in this study has removed this assumption by allowing the Poisson's Ratio to be calculated as part of the TSA. Primarily it is the prior estimate of the specific storage (also determined by TSA) which allows the Poisson's Ratio, along with all the other parameters, to vary within the theoretical ranges. As such, a relationship can be established between phase and amplitude in

the methodologies presented here. Within the post-strain model (Section 2.2.3, Hsieh et al. (1988)) a decrease in phase (larger negative number) acts to decrease both the hydraulic conductivity and specific storage, whereas for an increase in the amplitude response the hydraulic conductivity remains stable (marginally increases) and the specific storage decreases. This can be summarised as:

$$\phi \downarrow = k^f \downarrow , S_s \downarrow ; amp \uparrow = k^f \sim , S_s \downarrow \tag{36}$$

For the pre-strain model (Section 2.2.4, Wang (2000)) an increase in phase difference increases the hydraulic conductivity and decreases the specific storage, and an increase in the amplitude response increases the hydraulic conductivity but decreases the specific storage, summarised as:

$$\phi \uparrow = k^f \uparrow , S_s \downarrow ; amp \uparrow = k^f \uparrow , S_s \downarrow \tag{37}$$

Within the compressible grains model (Section 2.4.3), an increase in amplitude response results in an increase in the shear

modulus and a decrease in the porosity. Whereas an increase in barometric efficiency (i.e. smaller loading efficiency) results in an increase of both the shear modulus and porosity, summarised as:

$$amp \uparrow = G \uparrow , \theta \downarrow ; BE \uparrow = G \uparrow , \theta \uparrow \tag{38}$$

However, this is compounded with the change in the specific storage from either an increase or decrease in the amplitude response. Note the importance of this relationship and the effect of the $S_2$ disentanglement by Rau et al. (2020), where by

comparison previous methods which calculated BE such as Jacob (1939) or Acworth et al. (2017) overestimate the barometric loading. This in turn results in an overestimation of the shear modulus, which would also affect the derivations of other parameters according to Equation 20.

### 4.3 Strain responses reveal subsurface heterogeneity and anisotropy

Combining ET and AT responses in the subsurface analysis is based on the principle that Earth and atmospheric tides induce

strains with a different directionality. ET is fundamentally cubic, but is approximated as planar (tidal dilatation) (Schulze et al., 2000; Fuentes-Arreazola et al., 2018). However, Rojstaczer and Agnew (1989) stated that the use of the horizontal areal strain from Earth tides is a sufficient approximation for subsurface depths of up to thousands of kilometres. For ET, the strain is experienced in the vicinity of the well bore screen, although the distribution of this stain radially (cylindrical or spherical) from the screen is uncertain. The subsurface strain response to Earth tide induced stress depends on the elastic properties which are





highly heterogeneous on a small scale. However, the pore pressure response as measured by a well intersects a larger volume and should therefore be representative of the theoretical values derived from Earth tide calculations.

    Rojstaczer and Agnew (1989) predict that the response of ET (areal strain) should be high for low porosity and compressibility. Similarly, for such conditions, the barometric efficiency should approach one ($BE \rightarrow 1$, or equivalently $\gamma \rightarrow 0$). However, this does not necessarily occur as can be seen in our results for Death Valley and Dodowa where the groundwater response

magnitude to ET is large but BE is significantly smaller than unity. This phenomenon can be explained by the fact that BE is estimated vertically across a typical geological profile as a surface load, uni-axially compressing the subsurface. Here, consolidation generally increases with depth and we hypothesise that the AT response vertically integrates the material properties above the monitoring point, i.e. the result is representative of the vertical heterogeneity in elastic properties encountered. The precise geometry of the representative volume from either ET or AT is currently unknown, but it is assumed to be equivalent.

However, if this assumption is flawed and the representative volumes of ET or AT significantly differ, strain anisotropy may exist between these two forces and complicate their joint interpretation. Detailed field experimentation or coupled hydraulic-geomechanical modelling would be required to explore such a phenomenon.

### 4.4    In-situ conditions explain discrepancy in poroelastic properties

    Our results in Table 4 largely comply with previously established values (Wang, 2000), except for the observation of negative

Poisson's ratios. It is important to note that previous studies typically assume a literature value for Poisson's ratio when calculating geomechanical properties (Cutillo and Bredehoeft, 2011). Our new approach is enabled through tidal disentanglement to remove the need for such an assumption. However, the negative Poisson ratios are a surprising result and require explanation.

    It is theoretically possible for Poisson's ratio to range between negative one and positive half, i.e. $-1 \leq \nu \leq 0.5$ (Lakes, 1991; Lakes and Witt, 2002). Here, materials with a negative Poisson's ratio are described as auxetic, i.e. materials that become

thicker parallel to the direction of the stress. The occurrence of auxetic behaviour in rocks was discussed by Gercek (2007), who summarised that as a Poisson's ratio becomes increasingly negative ($\nu \rightarrow -1$), the material become highly resistant to shear deformations but easy to deform volumetrically. Ji et al. (2018) succinctly describe this relationship for conditions where the shear modulus is much greater than the bulk modulus, defined as $K < 2G/3$, and geologically is likely associated with highly anisotropic rocks. This ratio between the bulk and shear modulus is consistent with all results presented in this paper.

As such, the negative Poisson's ratios are indicative of the subsurface laterally contracting while being vertically compressed, following the theory of linear poroelasticity.

    Previous instances of negative Poisson's ratios for standard uniaxial core sample testing have been recorded by Homand-Etienne and Houpert (1989) and Zhao et al. (2020) in thermally induced micro-cracked granites. However, reporting of auxetic behaviour in rock is dominated by studies involving low strains and low confining pressures. For example, in the Berra Sand-

stone, Handin et al. (1963) observed that small compressive strains (here, small was defined as less than 200 Bar $\approx$ 2000 $mH_2O$ or 20 MPa) for confining pressure conditions cause the dilation of pore spaces. Comparatively, observations of pore volumes remained constant for moderate strains (20 to 50 MPa) and reduced in volume for large strains ($> 50$ MPa). Ji et al. (2018) have recently examined auxetic behaviour over a broad range of lithologies and pressures. They concluded that negative





Poisson's ratios are possible in crystalline igneous and metamorphic rocks (non-fractured) for confining hydrostatic pressures

less than 5 MPa, and less than 200–300 MPa for more quartz-rich sedimentary rocks such as silt stones and sand stones. Further, Ji et al. (2018) observed that the porosity of sedimentary rocks plays an important role in controlling auxetic effects, similar to the nano-scale fabric in artificial auxetic materials (e.g. metallic foams).

The results in this paper are obtained in-situ for fully saturated, undrained (confined) conditions and caused by small magnitude strains, which are conditions that differ considerably from those used in traditional laboratory techniques for determining

elastic moduli (i.e., $E$, $G$, $K$, $\nu$). Compared to the conditions experienced during a compressive laboratory test, or those described above by (Ji et al., 2018), the strains caused by EAT are very small. For example, the loading variations caused by the atmospheric tidal component $S_2$ is typically less than $9 \cdot 10^{-5}$ MPa (0.1 $mH_2O$), and the confining pressure caused by an artesian standing water level of 100 $mH_2O$ equates to a confining pressure of only 0.98 MPa. Laboratory results are also well known for demonstrating bias in the sample strength, with the strength decreasing with the sample's increasing physical

size. It has been found that this occurs due to the incorporation of heterogeneities in the sample at larger scales, such as minor lithological changes or discontinuities due to fracturing or jointing (Cundall et al., 2008; Masoumi et al., 2016).

Alternative subjective in-situ methods, such as seismic based methods (Rau et al., 2018), still derived positive Poisson's ratios when passing through the same heterogeneous material at the same confining pressures. However, elastic moduli have previously been shown to be frequency dependent when saturated and under confining pressure (Wang, 1993; Tutuncu et al.,

1998). Here, we hypothesise that the low frequency of the EAT induced stresses ( 2 cpd $\approx 2.3 \cdot 10^{-5}$ Hz), compared to seismic propagated waves (1 to 100 $Hz \approx$ 86,400 to 8,640,000 $cpd$), causes a highly relaxed response which allows sufficient time for pressure redistribution (Tutuncu et al., 1998). In contrast, the seismic frequency produces a localised un-relaxed or un-drained response as the seismic waves pass through the subsurface, where this effect has been shown to change with the frequency (Pimienta et al., 2016). Both (Tutuncu et al., 1998) and (Pimienta et al., 2016) provide evidence of decreasing Poisson's ratios

with decreasing frequency when below the typical undrained response domain ($< 10$ $Hz \approx 864,000$ $cpd$).

For small strains, as relevant for this study, Zaitsev et al. (2017) have shown that the occurrence of negative Poisson's ratios is not as exotic as previously thought. Considering the context of Cundall et al. (2008), Gercek (2007) and Ji et al. (2018), the negative Poisson's ratios derived by TSA in this paper seem plausible. Here, we propose that these are due to an interplay of simultaneous conditions for the in-situ determination, such as the scale of the effective sample size, anisotropic strain

responses from heterogeneities, low confining pressures, and the low frequency and small strains caused by EATs. Meeting the requirements of a negative Poisson's ratio at these small strains defined by (Lakes, 1991) as non-affine deformation (non-uniform between scales), non-central forces, and in a state of pre-existing strain (e.g., from overburden). The geomechanical derivations of this paper (Section 2.4) are based on linear poroelasticity. However, the auxetic responses observed by Ji et al. (2018) occurs both linearly and non-linearly within the negative Poisson's ratio space, depending on the confining pressure and

the type of material (Zaitsev et al., 2017). Currently, no relationships between EAT and nonlinear poroelastic theory has been established within the literature. Future work in this space should therefore consider the integration of nonlinear geomechanics (Khan et al., 1991; Johnson and Rasolofosaon, 1996).





To the best knowledge of the authors no explicit or robust relationships exist in the literature between elastic moduli results obtained in the field to those estimated from the laboratory testing of core (Leriche, 2017). Similarly, no in-situ method currently

exists that can derive elastic estimates of thousands of cubic meters of material (e.g., meters around a well bore screen), as has been proposed for Earth tides (Zhang et al., 2019). Over such a large volume, heterogeneity within almost any geological media will produce an anisotropic strain response to either Earth or atmospheric tides. Such anisotropy may result in apparently atypical properties, such as negative Poisson's ratios, and should be investigated for the generic assumption common to most hydro- or geomechanical investigations of a homogeneous, isotropic aquifer of infinite lateral extent.

### 610   4.5   Implications for passive quantification of subsurface hydro-geomechanical properties

There are several uncertainties associated with the findings of this paper, with implications for passive quantification of subsurface hydro-geomechanical properties. These uncertainties and limitations of the method are as follows:

– Although subjective estimates have been attempted (Zhang et al., 2019), the size and scale of the volume of influence from either ET or AT are unknown. It is also possible that there is a difference between the size of influence for ET and

AT. Further research is required to elucidate the zone of influence the derived properties are representative for, such as numeric modelling.

– Currently the poroelastic response to EAT is considered to be linear. However, rocks have previously been shown to respond in a nonlinear manner for undrained, tri-axially loaded laboratory settings, particularly at small strains (Johnson and Rasolofosaon, 1996; Zaitsev et al., 2017). As in-situ derivations of rock mass (or sediment) poroelastic values

without the use of assumed primary values ($E$, $G$, $K$, $\nu$) is relatively novel, the implication of assuming linearity for the analysis of in-situ properties remains unknown and unverified.

– The mechanism behind pre-strain responses is believed to be due to a partial drained response in the subsurface. However, the exact causes of such responses are still unknown. In order for the validity of a positive phase shift model to be proven, a more comprehensive understanding of such mechanisms must be further developed.

– Skin and well bore storage effects have been assumed to be negligible in this paper. However, these two effects will alter the phase responses to either Earth or atmospheric tides, as was shown in the recent work by Gao et al. (2020). It is important to note that any phase uncertainties mainly influence the hydraulic conductivity values. However, additional consideration of skin and well bore storage effects will increase the accuracy and confidence in results.

– We note that there is very little literature reporting values let alone ranges of grain compressibility for mineralogy other

than quartz, as has been discussed by Rau et al. (2018). Since this is the only real unknown, further work is required to elucidate the effect of grain compressibility uncertainty on the accuracy of the parameter estimation.

Passively characterising the subsurface with in-situ analytical measurements may change the way in which the confined subsurface is understood. For example, the possibility of auxetic behaviour of subsurface materials in undrained conditions





(i.e. hydraulically coupled) will have implications for assessing compaction from groundwater estimates, or the stability of
slopes and cuttings. Here, the low strain elastic estimates from TSA may provide a lower bounding end-member for plausible
ranges of properties. With further study, it may be possible to infer poroelastic properties at different confining pressures
and frequencies or to provide a more accurate in-situ determination of geomechanical rock properties (e.g. specific storage,
strength, etc.) prior to excavation and construction of civil and mining projects.

## 5   Conclusions

The method developed in this paper provides a comprehensive approach to estimate in-situ hydro-geomechanical properties
using Tidal Subsurface Analysis (TSA), i.e. from the monitored groundwater response to Earth and atmospheric tides (EAT).
Our new method first objectively disentangles the groundwater response to Earth tides (ET) and atmospheric tides (AT) for
the dominant response frequencies ($M_2$ and $S_2$). Secondly, the approach uses the amplitude and phase responses to ET and
AT to determine the complete hydro-geomechanical parameter space: Specific storage, hydraulic conductivity, porosity, shear,
Young's and bulk modulus, undrained and drained Poisson's Ratio, Skempton's and Biot-Willis coefficients. Unlike previous
research, our new approach does not require an a priori estimate of the Poisson's ratio. However, the application to consolidated
systems requires an estimate for the grain compressibility for which literature based values can be used.

Application of our new method to five groundwater and barometric pressure records from four different hydrogeological
settings delivers physically realistic results that are consistent with previous estimates. However, we reveal that the in-situ
estimates of Poisson's ratio are consistently negative indicating auxetic behaviour. A closer look at the literature reveals that
this is not unrealistic and can be attributed to an interplay between simultaneous in-situ conditions that differ from those of
established laboratory tests. These include a larger effective sample size with scaling effects, anisotropic strain responses due to
heterogeneities (e.g., micro-cracking), significantly lower confining pressures, and the small strains at low frequencies caused
by the EATs.

Our approach allows estimation of the complete hydro-geomechanical parameter space in a passive way, i.e. from monitoring
records of groundwater pressure head, measured atmospheric pressure and calculated ET. The primary advantage is that all
parameters are determined for the same in-situ conditions and that the estimated values therefore should be internally consistent.
The new method enables site-specific heterogeneity to be evaluated, as was shown by the two evaluated records from sandstone
bores, providing hydro-geomechanical properties of the rock mass rather than small scale estimates on intact rock. This is a
clear advantage to methods that require taking samples to the laboratory where replicating field conditions such as in-situ
confining pressure and representative scale can be problematic. However, our method also raises the need for further research
in key areas where significant uncertainties remain, for example the possibility for non-linearity of the poroelastic response to
surface loading and Earth tide forces. Addressing the identified uncertainties could contribute towards improving subsurface
monitoring and characterisation in both consolidated and unconsolidated systems.



*Code and data availability.*  Data and code will be made available in a repository if this work is accepted for publication.

*Author contributions.*  TCM and GCR conceived the idea for this paper. TCM and GCR analysed the data and made the figures. MSA and WAT contributed with datasets and suggested improvements.

*Competing interests.*  The authors declare no competing interests.

*Acknowledgements.*  Thanks are due to Francis Andorful, George Lutterodt, and Jan Willem Foppen from the T-GroUP project for drilling
observation well BH11, installing a diver, and permitting the use of the Dodowa groundwater hydrograph data. We thank Paula Cutillo and Shannon Mazzei from the National Park Service (NPS) in California (USA) for providing the barometric and groundwater pressure dataset for BLM-1. Some of the data used in this paper were collected with equipment provided by the Australian Federal Government financed *National Collaborative Research Infrastructure Strategy* (NCRIS). This project has received funding from the European Union's Horizon 2020 research and innovation programme under the Marie Skłodowska-Curie grant agreement No 835852. TCM was partly supported by an
Australian Government Research Training Program (RTP) Scholarship.



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
