# Peer review of "In-situ estimation of subsurface hydro-geomechanical properties using the groundwater response to semi-diurnal Earth and atmospheric tides"

_Hydrology and Earth System Sciences, 2021_

## Author Comment (AC1)

**Author response to review**

We thank the editor and the two anonymous reviewers for their constructive comments and suggestions. We thank RC1 for the review opinion that 'I support for publication of this work' and RC2 for the review opinion that 'I suggest this paper to be published after some revisions. We believe that in addressing the issues that these reviewers have raised, the paper will be considerably improved.

In the following we respond to the reviewers' comments. To facilitate easy assessment, we colour coded our responses into neutral (blue), agreement (green), partial agreement (yellow) and disagreement (red). We hope that our responses convince the editor to ask for revisions.

**RC1: 'Comment on hess-2021-359', Anonymous Referee #1, 07 Sep 2021**
**General comments on the methods:**

The methodology is quite articulated and includes the integration of several models and mathematical formulations. It was not easy to follow the presentation of the method. For example, two models are presented for post or pre-strain response. While after reading the whole manuscript I (probably) understood the reason for this, the way the models are integrated is only explained in section 2.2.5 after the models are presented in 2.2.3-4. There is an itemized list, presenting the whole method step by step, but this comes only in section 3. I suggest presenting a flowchart presenting the general idea of the whole methodology at the beginning of section 2, to clarify the explanation and presentation of the methods.

We agree. The method outline (currently Section 3.2) will be merged with the theoretical coverage of the previously developed methods (as also requested by RC2). We will add a flow chart to aid the reader's understanding.

To me it is not intuitive to understand the meaning of a pre-strain water response. After reading the whole manuscript I understand that the authors are offering here a quantitative interpretation to a phenomenon that is not clearly understood and, in these terms, I completely support their work. However, all the discussion related to this point, should be given earlier (see e.g., section 4.2 495-505) to give the reader the possibility to properly assess and understand all the assumptions. On this note, the caption of figure 3 distinguishes confined from semi-confined, and this is related to post and pre-strain as far as I understand. This is not clear from the figure caption.

We agree and will overhaul the discussion. We will change the terminology to reflect the physics of the system: Pre and post strain referred to the observed phenomena how a well water level response occurs in relation to the strain. In reality, the phase shift is caused by vertical water leakage through a semi-permeable zone. This shifts the harmonic so that the signal appears to occur earlier. We will rename "pre-strain response" to "leaky response", whereas "post-strain response" will become "confined response". Further, we propose to move the discussion how harmonic response results are compared to existing literature to Section 2 (Theoretical background). This provides better theoretical context before the results are presented.

**General comments on the results and discussion:**

In general, there is little appreciation in the paper for the uncertainty associated with the estimated quantities. I suggest reporting more details about parameter estimation via curve fitting, e.g., the value assumed by residuals, RMSE or other similar error indicators, and the confidence bounds associated with the key estimated properties.

We will do our best to report the results of uncertainties and correlations of the parameter estimation in our revised manuscript. Numerical uncertainties originate from the harmonic least-squares (HALS) routine. We will also try and propagate uncertainties through the non-linear solving routines.

Following up on this comment I have a particular concern: the method prescribes the selection of pre- or post-strain model according to a phase shift evaluation, where the two models are assumed to be both possible if the shift is between -1deg and 0deg. However, I wonder if this overlap is not too restricted as, in principle, the estimation of the phase shift from observed data may be affected by a larger interval of uncertainty.

We agree that this is an issue. However, uncertainty in phase estimates predominantly affect the estimated hydraulic conductivity. This parameter is not subsequently used when further properties are estimated. Therefore, it should not be an issue. We are going to note this in our revised manuscript.

I found the discussion of the results quite interesting, particularly the fact that they seem to disclose nontrivial response of subsurface materials to EAT. These could be due to multiple factors, as extensively discussed in the paper. I have three comments on this point, which in my view should be considered in a revision:

I wonder if the observed results may be the results of some particular assumption embedded in the parameter estimation procedure. In particular, can the authors demonstrate that the proposed methodology leads to consistent results when applied to synthetic data (i.e. data numerically generated with known parameters and artificially perturbed)? This would demonstrate that the estimation method is robust in terms of parameter identification, at least when the data are consistent with the assumptions.

We are unsure what "numerically generated" refers to in this context. If the reviewer refers to the parameter estimation, we applied two different checks in our work: (1) we tested the accuracy and reliability of the signal processing using synthetically created harmonics and Gaussian noise (Schweizer et al., 2021); (2) we tested the non-linear solver for delivering a unique solution. We will note these points in the revised manuscript. If the reviewer means a numerical model of the physical subsurface processes, then this would be out of scope for the current manuscript.

Regarding the discussion of the negative Poisson ratio (section 4.4), and given the fact that the measurement sampling volume is unknown, is it possible that this result is due to boundary effects?

Yes, this is possible. However, we cannot conclusively answer this as there is a lack of knowledge in the literature. We will include this as a possible cause in our revised discussion.

The authors state that the results could be used to infer poroelastic properties to be used in civil and mining construction. However, I have the impression that some of the estimated parameters may be driven by the very specific conditions associated with EAT, and may not be portable to different conditions and loading. For instance, I wonder if some of the observed parameters may be associated with different time scales associated with material responses.

We appreciate this comment. There is a lack of general understanding of the portability of EAT derived results, including their frequency dependency. We will revise our discussion to clarify this further.

In section 4.2 the authors provide a sort of sensitivity analysis. (e.g., eq. 36-37). This is hard to follow because the discussion is only qualitative. I suggest either dropping it or expanding it. However, the paper is already dense and long and I wonder if the author really need to include this point.

We agree with the reviewer that expanding upon this would further lengthen the work. Rather, this should be addressed in a future contribution. Also, we believe it is useful that we mentioned our checks mentioned earlier. We will add a statement to the revised manuscript that this needs further work.

**Other minor points:**

Data Figure 3 are scarcely readable, please improve readability of the Figure.

Figure 3 will be improved according to the suggestion.

Line 405: if bounds are imposed it seems quite logical that no none of the parameters exceeded the fitting bounds. I suggest rephrasing and, as mentioned above, provide more quantitative details about the estimates.

The intent was to illustrate that the results were not biased by fitting problems. We will revise this and add more quantitative information about the estimates.

At line 648 the authors state that the model offers the advantage to rely on information on grain compressibility, available in the literature. However, at line 629 they state the opposite, i.e. that grain compressibility data are generally lacking. Please reconcile the two statements.

What we meant was that grain compressibility can be considered, if available. However, to the best of our knowledge, the only available data in the literature is from quartz. Consequently, our application was limited to using this. We will reconcile these statements to better reflect this.

---

## Author Comment (AC2)

**Author response to review**

We thank the editor and the two anonymous reviewers for their constructive comments and suggestions. We thank RC1 for the review opinion that 'I support for publication of this work' and RC2 for the review opinion that 'I suggest this paper to be published after some revisions. We believe that in addressing the issues that these reviewers have raised, the paper will be considerably improved.

In the following we respond to the reviewers' comments. To facilitate easy assessment, we colour coded our responses into neutral (blue), agreement (green), partial agreement (yellow) and disagreement (red). We hope that our responses convince the editor to ask for revisions.

**RC2: 'Comment on hess-2021-359', Anonymous Referee #2, 10 Nov 2021**
**General comments**

The paper is very long and organization of the sections is quite confusing. The methodology is described only on page 19, even after the description of the 4 studied sites. In order to better understand the sections on the pre- and post-strain response of the water level response and the discussions on the compressibility or incompressibility of grains, the methodology should be introduced right after the introduction.
We will revise the methodology in line with this and comments by RC1.

In the different stages of the methodology, the hypotheses behind the employed theories should appear more distinguishly: drained or undrained conditions, consolidated or unconsolidated, lateral flows or only vertical flow etc.
We will add the specific assumptions behind the theory to the methodology.

Finally, the title should be more precise since the groundwater response was studied only at semi-diurnal periods (for instance add "semi-diurnal" before Earth and atmospheric tides) and precise M2 and S2 tides in the abstract. The surface load model used in the pre-strain water level response and the model used in the post-strain response are frequency-dependent, elastic parameters too. We may expect different results when analyzing diurnal or longer-period tides for instance.
We agree and will revise the title accordingly.

**Detailed Remarks**
p.2 line 45: please define ETs
ET will be defined.

p.2 line 46: "separating tidal components" it depends mostly on the spectral resolution, hence on the length of the time-records used.
This statement is true for the Fast Fourier Transform (FFT), but it is different for harmonic least-squares (HALS) which is what we used. For further information please refer to the recent work by Schweizer et al. (2021). We will add a clarifying statement to the manuscript.

p.3 line 71: You applied a moving average spanning across a time period of 3 days; such a process is equivalent to a low-pass filtering not high-pass filtering. It filters out higher frequency signals. Please replace "longer frequency" by "higher", since I do not know what means a longer frequency.
We will revise this mistake.

p.3 section 2.1: how well are identified the M2 and S2 tidal components in the data using HALS? How large are the uncertainties on the amplitudes and phases? Particularly on the phase, how precise it is, since it will affect the phase-shift value used to determine the use of a pre-strain or post-strain model.
This comment aligns with the uncertainty analysis suggested by RC1. We refer to the response given earlier.

p.5 equation (2): the superscript "p" in the following equations designs "pore" but here is this "p" for potential? Please clarify.
In this context, "p" is not a subscript. We acknowledge that this is hard to distinguish and will change all "ETp" to "ET_pot".

p.6 line 138 typo: dilation à dilatation
This will be corrected.

p.7 section 2.2.3 the figure 3 is referenced here before Figure 2. Please reorganize figures in order they are numerated in the order of citation.
This will be corrected.

p.8 section 2.2.3 some discussion on the boundary layer depth associated with the parameter aw/rw should be done in regard with the pre-strain model depth here after.
We are confused as the reviewer refers to section 2.2.3 but also a "boundary layer depth" and pre-strain model (which is section 2.2.4). We are unsure what this means.

p.8 section 2.2.4 more discussion on the boundary layer depth is missing. For instance, at which depth/diffusivity the amplitude AM2 is maximum?
The depth/diffusivity combination at which A_M2 is maximal can be seen in Figure 2a. We will add a statement including a reference to the figure.

Interpretation of Fig. 2 is missing. For instance, with respect to the plots shown in Fig. 6.11 in the book by Wang (2000), at what depth the diffusive pore-pressure effects are confined? What is the limit in terms of thickness for using this theory as a good approximation? What about the phase of eq. (14), if you plot it wrt z/d, in which depth/diffusivity range does the sign change?
XXX

p.8 line 196 is this 10 m the value obtained for d when the pore-pressure is equal to surface load? How much larger the pore-pressure can be wrt surface load (when loading efficiency is larger than 1)? Please explain better the adequacy (the valid depths ranges) when combining ET and AT.
We will add a statement to clarify this.

p.10 section 2.3 please introduce here BE = barometric efficiency. This section related to damping could be put into or right after section 2.2.2.
We decided to separate the influences of ET and AT in the methodology as they require disentangling before combining the obtained results. Section 2.3 discusses the influence of AT and moving this after 2.2.2 would merge this with the section discussing ET influences. This change would confuse the reader's understanding of the separate mechanisms that are shown in Figure 1.

p.12 equation (26) in the denominator, the \theta should be rather a \gamma.
This mistake will be corrected.

p.14 equations 32-35 are solved using an iterative LS scheme. Why not using a Bayesian inference in particularly to check the correlations between the various parameters?
We will use a more sophisticated approach to obtain uncertainties and correlations between the various parameters.

p.17 line 349 please define MASL (m above sea level)
We will revise as suggested.

*p.17 last line: "the ~28 days" as the minimum requirement for what? In order to separate M2 and S2 in terms of frequency resolution we would need 57 days. Please precise.*
We wish to point out that the reviewer is likely referring to the Fast Fourier Transform. In this work we use harmonic least-squares (HALS) which has been tested by Schweizer et al. (2021). We will clarify this including a citation to avoid confusion.

*p.18 line 378 Detrending using SciPy function detrend is done by fitting a linear function, not by moving average, please correct this sentence; the moving average enables to low pass filter the data.*

We agree and will revise this.

p.21 section 3.3 The choice of the post-strain model for the Death Valley site should be discussed since the phase shift of -1 degree is at the limit between pre and post-strain models.

This dataset was previously analysed by Rau et al. (2020), who justified the use of their model choice. We will clarify this by adding a reference.

p.24 section 4.2 I do not really understand this long discussion. It should be simplified in order to highlight the major points.

We will simplify this section as appropriate.

p.24 line 497 typo: stain à strain

This mistake will be corrected.

p.25 line 533 typo stain à strain

This mistake will be corrected.

p.28 lines 606-609 these statements have already been claimed before, please remove this repetition.

We will review this and revise as appropriate.

p.26 section 4.4, discussion on the negative Poisson ratio. What about the influence of ocean loading? Have you quantified its impact on the amplitudes and phases of M2 and S2 for the 4 sites considered in this study? Uncertainties on the M2 and S2 phases should be discussed too since it may influence the values of the Poisson ratios obtained at the end. Correlation between the parameters should be checked too.

We have not investigated the influence of ocean tide loading on the M2 and S2 components. We will do this for the locations given and assess how this would influence our results. We have responded to the request for uncertainty as part of RC1.

---

## Author Response (AR1)

**RC1: 'Comment on hess-2021-359', Anonymous Referee #1**

We thank the editor and the two anonymous reviewers for their constructive comments and suggestions. We thank RC1 for the review opinion that 'I support for publication of this work' and RC2 for the review opinion that 'I suggest this paper to be published after some revisions'.

Please note the following:
- We have made changes to the authorship to reflect the overall intellectual contribution and work including these revisions.
- To shorten the manuscript, we decided to drop the former Table 2 and add a reference to where the mathematical relationships can be found in the literature. The methodology section already contains all required equations.
- We have also decided to separate the results into two tables instead of one, where hydraulic estimates are now in Table 3 and poroelastic estimates are now in Table 4. This should improve readability of the manuscript and interpretability of the results.
- We have made many minor tweaks and improvements to the manuscript text to address the reviewer's suggestions (see track changes).

To facilitate easy assessment, we colour coded our responses into neutral (blue), agreement (green), partial agreement (yellow) and disagreement (red). We hope that our responses convince the editor to ask for revisions. We believe that we managed to significantly improve this paper in response to the review.

**General comments on the methods:**
The methodology is quite articulated and includes the integration of several models and mathematical formulations. It was not easy to follow the presentation of the method. For example, two models are presented for post or pre-strain response. While after reading the whole manuscript I (probably) understood the reason for this, the way the models are integrated is only explained in section 2.2.5 after the models are presented in 2.2.3-4. There is an itemized list, presenting the whole method step by step, but this comes only in section 3. I suggest presenting a flowchart presenting the general idea of the whole methodology at the beginning of section 2, to clarify the explanation and presentation of the methods.
We agree. The method outline (currently Section 3.2) has been merged with the theoretical coverage of the previously developed methods (as also requested by RC2). We believe that this has clarified the workflow.

To me it is not intuitive to understand the meaning of a pre-strain water response. After reading the whole manuscript I understand that the authors are offering here a quantitative interpretation to a phenomenon that is not clearly understood and, in these terms, I completely support their work. However, all the discussion related to this point, should be given earlier (see e.g., section 4.2 495-505) to give the reader the possibility to properly assess and understand all the assumptions. On this note, the caption of figure 3 distinguishes confined from semi-confined, and this is related to post and pre-strain as far as I understand. This is not clear from the figure caption.
We agree and have revised the manuscript. We have changed the terminology to reflect the physics of the system: Pre and post strain referred to the observed phenomena how a well water level response occurs in relation to the strain. We have changed "pre-strain response" to "leaky response", whereas "post-strain response" was changed to "confined response". Further, we have moved the discussion about harmonic responses to Section 2 (Theoretical background). This provides better theoretical context before the results are presented.

**General comments on the results and discussion:**
In general, there is little appreciation in the paper for the uncertainty associated with the estimated quantities. I suggest reporting more details about parameter estimation via curve fitting, e.g., the value assumed by residuals, RMSE or other similar error indicators, and the confidence bounds associated with the key estimated properties.
We agree that this is not addressed well. Please note that our mention of curve fitting in the manuscript was misleading. We solve systems of non-linear equations using root finding because they cannot be solved explicitly. The root finding uses least-squares minimisation, and the iteration is set to stop when the numerical errors reach machine precision (i.e., virtually no uncertainty). Note that this approach is not the

same as parameter estimation when fitting a function to a dataset, i.e., normal curve fitting. Root finding itself does not lead to considerable uncertainty in the estimates. The only possible origin of method-based uncertainty is HALS which was comprehensively tested in Schweizer et al. (2021). We further note that errors originating from conceptual model assumptions inherent to the analytical solutions or boundary effects (see comment below) are potentially much higher and more important than those from solving non-linear equations. We have reformulated our misleading statements (e.g., the use of the term "curve fitting") and made other changes to the manuscript to clarify this as follows:

"*Schweizer et al. (2021) further noted that HALS outperforms the discrete Fourier transform, but also that devising an objective error estimation for HALS is difficult, as it depends on the nature of the residuals (difference between measurement and model), and this requires separate investigation.*" (lines 447-)

"*This points to possible issues with simplified conceptual models and the validity of their assumptions. Further research is required to test the applicability of analytical solutions based on simplified assumptions applied to real-world conditions.*" (lines 489-).

Following up on this comment I have a particular concern: the method prescribes the selection of pre- or post-strain model according to a phase shift evaluation, where the two models are assumed to be both possible if the shift is between -1deg and 0deg. However, I wonder if this overlap is not too restricted as, in principle, the estimation of the phase shift from observed data may be affected by a larger interval of uncertainty.

Please note that the phase shifts for BH11 and Thirlmere 2 (see Table 3) are outside that range and fall squarely within the range of the confined model. Figure 3b shows that this only affects the hydraulic conductivity and not the specific storage. Further, neither phase shift nor hydraulic conductivity are parameters subsequently used for poroelastic properties estimation. These examples illustrate that phase errors should not be an issue.

I found the discussion of the results quite interesting, particularly the fact that they seem to disclose nontrivial response of subsurface materials to EAT. These could be due to multiple factors, as extensively discussed in the paper. I have three comments on this point, which in my view should be considered in a revision:
I wonder if the observed results may be the results of some particular assumption embedded in the parameter estimation procedure. In particular, can the authors demonstrate that the proposed methodology leads to consistent results when applied to synthetic data (i.e. data numerically generated with known parameters and artificially perturbed)? This would demonstrate that the estimation method is robust in terms of parameter identification, at least when the data are consistent with the assumptions.
Thank you for your interest in our discussion. We are unsure what "numerically generated" refers to in this context. If the reviewer means a numerical model of the physical subsurface processes, then this would be out of scope for the current manuscript. If the reviewer refers to the parameter estimation, we applied two different checks in our work: (1) the accuracy and reliability of the signal processing method HALS was previously tested using synthetically created harmonics and Gaussian noise (Schweizer et al., 2021); (2) we comprehensively tested the non-linear root finding approach for delivering a unique solution without reaching or exceeding physical sensible bounds. We have noted these points in the manuscript. As mentioned earlier, we believe that the conceptual model is the main contributor to the observed results and that incorrect simplifying assumptions related to the conceptual model are likely to be the largest source of uncertainty. Interestingly, these simplifying assumptions typically also apply to numerical models of subsurface processes. We have revised the discussion and added further statements to reflect this:

"*This points to possible issues with simplified conceptual models and the validity of their assumptions. Further research is required to test the applicability of analytical solutions based on simplified assumptions applied to real-world conditions.*" (lines 489-).

Regarding the discussion of the negative Poisson ratio (section 4.4), and given the fact that the measurement sampling volume is unknown, is it possible that this result is due to boundary effects?
Yes, this is possible. However, we cannot conclusively answer this as there is a lack of knowledge in the literature. We have revised the discussion to include this as a possible cause:

*"We propose the following possible reasons which could lead to negative Poisson ratios, such as the scale of the effective sample size, anisotropic strain responses from heterogeneities, low confining pressures, the low frequency and small strains caused by EATs, and boundary effects."* (line 560-).

The authors state that the results could be used to infer poroelastic properties to be used in civil and mining construction. However, I have the impression that some of the estimated parameters may be driven by the very specific conditions associated with EAT, and may not be portable to different conditions and loading. For instance, I wonder if some of the observed parameters may be associated with different time scales associated with material responses.

There is a lack of general understanding of the transferability of EAT derived results, including their frequency dependency. We have tweaked the discussion (see track changes) and added the following statement:

*"However, further research is required to elucidate the scope of validity (space and time) and transferability of hydro-geomechanical properties derived from different methods."* (line 606-)

In section 4.2 the authors provide a sort of sensitivity analysis. (e.g., eq. 36-37). This is hard to follow because the discussion is only qualitative. I suggest either dropping it or expanding it. However, the paper is already dense and long and I wonder if the author really need to include this point.

We agree with the reviewer that expanding upon this would substantially exceed length limits. We have deleted this part of the discussion.

**Other minor points:**
Data Figure 3 are scarcely readable, please improve readability of the Figure.
We have improved the readability of Figure 3 by increasing the font sizes.

Line 405: if bounds are imposed it seems quite logical that no none of the parameters exceeded the fitting bounds. I suggest rephrasing and, as mentioned above, provide more quantitative details about the estimates.

The intent was to illustrate that the results were not biased by root finding problems. See earlier reply regarding parameter uncertainties. We revised this statement as follows:

*"(…) none of the estimated parameters reached or exceeded any of the prescribed solution constraints."* (line 342).

At line 648 the authors state that the model offers the advantage to rely on information on grain compressibility, available in the literature. However, at line 629 they state the opposite, i.e. that grain compressibility data are generally lacking. Please reconcile the two statements.

What we meant was that grain compressibility can be considered, if available. However, to the best of our knowledge, the only available data in the literature is from quartz. Consequently, our application was limited to using this. We have revised this as:

*"However, the application to consolidated systems requires an estimate for the grain compressibility for which literature-based values can be considered if available."* (line 615).
* * *
**RC2: 'Comment on hess-2021-359', Anonymous Referee #2**
**General comments**
The paper is very long and organization of the sections is quite confusing. The methodology is described only on page 19, even after the description of the 4 studied sites. In order to better understand the sections on the pre- and post-strain response of the water level response and the discussions on the compressibility or incompressibility of grains, the methodology should be introduced right after the introduction.

We have revised the methodology in response to similar comments by RC1 (see earlier details). We believe that the current comment is already addressed through these changes.

In the different stages of the methodology, the hypotheses behind the employed theories should appear more distinguishly: drained or undrained conditions, consolidated or unconsolidated, lateral flows or only vertical flow etc.

To better clarify theory applicability in relation to system consolidation, we renamed Section 2.4.2. and 2.4.3 to "Unconsolidated systems" instead of "Incompressible grains" and "Consolidated systems" instead of "Compressible grains". We have further modified the start of each section to clarify this (see track changes).

We have made small revisions to the methodology to clarify flow directions related to the theory (see track changes).

We also note that whether drained or undrained theory applies is already distinguished in the equations and parameters by presence or absence of the superscript "u" (for undrained conditions). We have added a statement as follows:

"*In the following, undrained parameters are depicted by the superscript $u$.*" (line 259).

Finally, the title should be more precise since the groundwater response was studied only at semi-diurnal periods (for instance add "semi-diurnal" before Earth and atmospheric tides) and precise M2 and S2 tides in the abstract. The surface load model used in the pre-strain water level response and the model used in the post-strain response are frequency-dependent, elastic parameters too. We may expect different results when analyzing diurnal or longer-period tides for instance.

We agree and have revised the title accordingly:

"*In-situ estimation of subsurface hydro-geomechanical properties using the groundwater response to semi-diurnal Earth and atmospheric tides.*"

**Detailed Remarks**

p.2 line 45: please define ETs

ET has now been defined.

p.2 line 46: "separating tidal components" it depends mostly on the spectral resolution, hence on the length of the time-records used.

This statement is true for the Fast Fourier Transform (FFT), but it is different for harmonic least-squares (HALS) which is what we used. For further information please refer to the recent work by Schweizer et al. (2021). We have revised this as follows:

"Recent work by Rau et al. (2020) compares methods that estimate amplitudes and phases from monitoring datasets and concludes that harmonic least squares (HALS) is superior compared to Fast Fourier Transforms (FFT)."

p.3 line 71: You applied a moving average spanning across a time period of 3 days; such a process is equivalent to a low-pass filtering not high-pass filtering. It filters out higher frequency signals. Please replace "longer frequency" by "higher", since I do not know what means a longer frequency.

We have revised this mistake.

p.3 section 2.1: how well are identified the M2 and S2 tidal components in the data using HALS? How large are the uncertainties on the amplitudes and phases? Particularly on the phase, how precise it is, since it will affect the phase-shift value used to determine the use of a pre-strain or post-strain model.

This comment aligns with earlier comments by RC1. Uncertainties for HALS are difficult to estimate and require separate work. We refer to the response given earlier.

p.5 equation (2): the superscript "p" in the following equations designs "pore" but here is this "p" for potential? Please clarify.

In this context, "p" is not a subscript. We have changed all "ETp" to "ET^pot" to clarify this.

p.6 line 138 typo: dilation à dilatation

Corrected.

p.7 section 2.2.3 the figure 3 is referenced here before Figure 2. Please reorganize figures in order they are numerated in the order of citation.
We have revised this so that Figures 2 and 3 are referenced in the correct order.

p.8 section 2.2.3 some discussion on the boundary layer depth associated with the parameter aw/rw should be done in regard with the pre-strain model depth here after.
We are confused as the reviewer refers to section 2.2.3 but also a "boundary layer depth" and pre-strain model (which is section 2.2.4). We are unsure what this means in the provided context.

p.8 section 2.2.4 more discussion on the boundary layer depth is missing. For instance, at which depth/diffusivity the amplitude AM2 is maximum?
The depth/diffusivity combination at which $A\_{M2}$ is maximal can clearly be seen in Figure 2a.

Interpretation of Fig. 2 is missing. For instance, with respect to the plots shown in Fig. 6.11 in the book by Wang (2000), at what depth the diffusive pore-pressure effects are confined? What is the limit in terms of thickness for using this theory as a good approximation? What about the phase of eq. (14), if you plot it wrt z/d, in which depth/diffusivity range does the sign change?
Confinement is given when the amplitude response to forcing remains high. This depends on the hydraulic diffusivity and depth, as seen in Figure 2.
We are unsure what "(…) limit in terms of thickness (…)" refers to? This theory applies to systems that comply with the assumptions of the analytical solution, regardless of the thickness.
The phase of Equation 14 can turn negative but only with small values. This is already highlighted in Figure 2b (blue coloured lines).
We have extended the depth of Figure 2 so that the field sites we used later are within the range.

p.8 line 196 is this 10 m the value obtained for d when the pore-pressure is equal to surface load? How much larger the pore-pressure can be wrt surface load (when loading efficiency is larger than 1)? Please explain better the adequacy (the valid depths ranges) when combining ET and AT.
We recognise that this paragraph was misplaced and misleading. We have moved some of it into Section 4.2 (lines 478-) where the applicability of this model is already discussed. Please note that the amplitude damping is considered when calculating BE (Equation 17). The statement relating to this was moved to Section 2.3 (line 222-) where it is more appropriate. We have added a better explanation of the solution space. This now answers the questions regarding pore pressure responses. For example, the pore pressure can be 1.069 times bigger than the surface load, as can be seen in Figure 2a. We hope that this addresses the concerns.

p.10 section 2.3 please introduce here BE = barometric efficiency. This section related to damping could be put into or right after section 2.2.2.
Section 2.3 already introduces barometric efficiency (BE). We decided to separate the influences of ET and AT in the methodology as their effects require disentangling before their theories can be combined in a solution. Section 2.3 discusses the influence of AT and moving this after 2.2.2 would merge this with the section discussing ET influences. The requested change would confuse the reader's understanding of the separate mechanisms that are shown in Figure 1.

p.12 equation (26) in the denominator, the \theta should be rather a \gamma.
We have corrected this mistake and made a few other small changes to the text related to this (see track changes).

p.14 equations 32-35 are solved using an iterative LS scheme. Why not using a Bayesian inference in particularly to check the correlations between the various parameters?
As explained in response to RC1, solving these equations is a purely numerical procedure and not curve fitting, i.e., this does not introduce any uncertainty. Therefore, we see no justification for using a Bayesian inference approach. Further, since it is not a curve fitting exercise, correlations between parameters do not matter.

p.17 line 349 please define MASL (m above sea level)
Revised.

p.17 last line: "the ~28 days" as the minimum requirement for what? In order to separate M2 and S2 in terms of frequency resolution we would need 57 days. Please precise.
We point out that the reviewer is likely referring to the requirements for Fast Fourier Transformation. In this work we use harmonic least-squares (HALS) which has been tested by Schweizer et al. (2021). To clarify, we have revised as follows:

"(…) for at least three months which is longer than the $\sim$28 days being suggested as the minimum requirement of the selected HALS method (Schweizer et al., 2021)." (line 331-).

p.18 line 378 Detrending using SciPy function detrend is done by fitting a linear function, not by moving average, please correct this sentence; the moving average enables to low pass filter the data.
Corrected.

p.21 section 3.3 The choice of the post-strain model for the Death Valley site should be discussed since the phase shift of -1 degree is at the limit between pre and post-strain models.
This dataset was previously analysed by Rau et al. (2020), who justified the use of their model choice. Further, given the depth of screen at this site (~600 m), it is unlikely that the system is not confined.

p.24 section 4.2 I do not really understand this long discussion. It should be simplified in order to highlight the major points.
We have simplified this already in response to a similar comment by RC1.

p.24 line 497 typo: stain à strain
Corrected.

p.25 line 533 typo stain à strain
Corrected.

p.28 lines 606-609 these statements have already been claimed before, please remove this repetition.
We have revised this in response to RC1.

p.26 section 4.4, discussion on the negative Poisson ratio. What about the influence of ocean loading? Have you quantified its impact on the amplitudes and phases of M2 and S2 for the 4 sites considered in this study? Uncertainties on the M2 and S2 phases should be discussed too since it may influence the values of the Poisson ratios obtained at the end. Correlation between the parameters should be checked too.
We note that ocean tide loading is a complicated sub-discipline of geophysics and comprehensively covering this would exceed the scope of this manuscript. Nevertheless, to get an idea of the potential errors, we contacted Dr. Hans-Georg Scherneck from the Dep. of Space, Earth and Environment at Chalmers University of Technology in Sweden. He has established the ocean tide loading web provider available online at: http://holt.oso.chalmers.se/loading/. Following some discussion, we send him the geo-coordinates of our sites and he used one of the most recent ocean tide loading models (FES2014b) to calculate the ocean tide loading strain for the M2 component. When compared with our outputs, these indicate a maximum possible magnitude of approx. 12-19% compared to the Earth tide strain calculated using ETERNA. We have used this to recalculate the hydraulic and poroealstic results, but this does not turn Poisson's ratios positive.

We have added the following information to clarify these points in the manuscript:
- We added the following statement to the methodology:
  "We note that these programs do not account for ocean tide loading, i.e., the deformation of the subsurface due to the weight of the ocean tides. Ocean tide loading causes harmonic subsurface strain that is added to that imposed by Earth tides. The actual subsurface strain amplitude variation depends on the phase of both signals and is, in the worst cases, either added or subtracted from

*the Earth tide. To understand the potential impact of this effect we used the ocean tide loading provider by Chalmers University of Technology (\url{http://holt.oso.chalmers.se/loading/index.html}) to estimate the $M_2$ aerial strain ($A^{OTe}_{M_2}$) for our five locations with a state-of-the-art finite element model (FES2014b). However, we further note that ocean tide loading is a complicated phenomenon (Jentzsch et al., 1997) and its detailed assessment is beyond the scope of this manuscript.*" (lines 120-)

- We revised the following statement in the results:
  "*Geo-position of the boreholes, theoretical Earth tide strain for the same duration and sampling frequency of each site as well as aerial strain amplitudes from ocean tide loading for $M_2$ are summarised in Table \ref{tab:inputs}.*" (lines 332-)

- M2 amplitudes for ocean tide loading for all sites were added to Table 3.

- We added the following statement to the discussion:

  "*We investigated whether the influence from ocean tide loading could be responsible. However, considering the maximum possible influence from ocean tide loading for each site did not lead to positive Poisson's ratios.*" (lines 518-).

- We added the following statement to the acknowledgements:
  "*We thank Hans-Georg Scherneck and Machiel Bos from Chalmers University of Technology for providing the ocean tide loading strains used in this manuscript.*"

Please also note our responses to RC1 regarding uncertainties and numerical solving.

---

## Referee Report (RR1)

Review on "**In-situ estimation of subsurface hydro-geomechanical properties using the groundwater response to semi-diurnal Earth and atmospheric tides**"

by Gabriel C. Rau, Timothy C. McMillan, Martin S. Andersen and Wendy A. Timms

Summary

I thank the authors for performing major revisions in the structure and text of the manuscript that make it clearer and more pleasant to read. I also thank them for responding to all my remarks with clear answers. This paper is now undoubtedly worth publishing in HESS after the authors perform some minor corrections listed hereafter.

Detailed remarks

p. 5 line 99 equation (1): the definition of the tidal potential V is missing and should be connected to the incomplete sentence "where M2 is the tidal frequency"

p. 6 line 107: when computing theoretical tides, for instance with *predict.for* function within ETERNA software, a model is used for the Earth. This model can be elastic or anelastic. In the later case a phase is introduced in the tidal constituents. In your tidal prediction using *PyGTide*, global anelasticity of the Earth should be considered to have correct M2 and S2 theoretical phases. Please provide some additional information on the Earth's model used for the tidal predictions.

p. 10 line 202: "(e.g., -0.5°C in Figure 3b)" Please remove the "C" since this is degree not degree Celsius.

p. 10 lines 202-203: "Figure 2c,d show the solution space when considering the strain response as well as separation of hydraulic properties" This sentence seems unconnected to previous ones and to Fig. 3. Please clarify this remark by adding for instance "at leaky conditions".

p. 19 Figure 4: it is strange to see pressure in *m* instead of *Pa*. Maybe add in the legend something like "barometric pressures (in equivalent water heights in *m*)"

References: please correct references for repeated url or doi.

---

## Author Response (AR2)

**Response to Review iteration 2 for HESS Manuscript 'hess-2021-359':**

We thank the editor and the two reviewers for their constructive comments and suggestions. We have addressed all points and provide detailed answers below. We hope that these revisions allow our manuscript to be published in HESS.

**Report #1**
**Suggestions for revision or reasons for rejection**
The authors have provided a point-by-point response to my comments. In some cases the comments have resolved the comments, for example I appreciate the new terminology for leaky response which I find much easier to follow. However, I do have a few remarks mainly concerning the quality of the text and the presentation of the work. I have to add that I got confused in the review because the version with marked changes does not seem to correspond to the last manuscript version (or at least it is not fully consistent). This should be double-checked. Below a list of points that need to be addressed, for which I am referring to the file manuscript-version-3.
We tried our best to make a track changes document in the LaTeX environment, which was not easy given all the revisions that were done. We apologise for any confusion this may have caused.

It is clear now that the methodology provides a deterministic analysis leading to the calculation of flow and poro-elastic parameters from observed data and the issue of parametric uncertainty is actually not considered in the paper. In light of this comment, I find misleading the title of section 4.1 which includes the expression "parameter estimation". To me this sounds like a statistical estimation method, but this is not what is meant here. I suggest changing the title of this section. Also the text includes a discussion about uncertainties which remains unresolved and I am not sure that it makes sense to include it, as it might be misleading (as it was for me).
We have renamed the title for section 4.1 to "Influences on the quantification of properties".
In terms of discussion about uncertainties, we believe that the reviewer is referring to this sentence:

*"Schweizer et al. (2021) further noted that HALS outperforms the discrete Fourier transform, but also that devising an objective error estimation for HALS is difficult, as it depends on the nature of the residuals (difference between measurement and model), and this requires further investigation."*

We wish to leave this standing, as it points to the need for further research. We believe that clearly pointing out the remaining knowledge gaps and future research needs, and directions are important aspects of a good Discussion section.

Line 490: "Further research is required to test the applicability of analytical solutions based on simplified assumptions applied to real-world conditions." The authors have added this sentence to reply to one of my comments. However, I thought the point of the paper was precisely to prove the applicability of analytical solutions to real world test cases. So I find the statement confusing and not properly addressing the point. This response should be revised again in my view.
We have reformulated this as follows:

*"Further research is required to independently validate results derived from passive methods that are based on simplified conceptual understanding and their analytical solutions and to test the influence of different and more complex real-world conditions, such as geological heterogeneity at different scales."*

We hope that this clarifies what we mean.

Conclusions: The authors state that "The new method enables site-specific heterogeneity to be evaluated …", but I do not see how. The method gives a unique estimate of the parameters (as it is now clarified), there is no way to determine any spatial variability, and the support scale of the measurement is unknown. The parameters probably correspond to some "effective" properties, but I am puzzled when I try to imagine how these measurements can be used to characterize heterogeneous fields of aquifer properties.
We have revised the paragraph as follows:

*"Our approach allows estimation of the complete hydro-geomechanical parameter space in a passive way, i.e. from monitoring records of groundwater pressure head, measured atmospheric pressure and calculated ET. The primary advantage is that all parameters are determined for the same in-situ conditions and that the estimated values therefore should be internally consistent. The new method provides hydro-geomechanical properties of the larger rock mass. This is a clear advantage to methods that require taking samples to the laboratory where replicating field conditions such as in-situ confining pressure and representative scale can be problematic. When combined with laboratory estimates on intact rock, it enables evaluation of scale-specific heterogeneity. Further, our method enables more monitoring bores to be tested for hydro-geomechanical properties at a lower cost compared to conventional aquifer pump testing. There is thus the possibility of better characterizing the heterogeneity of aquifer properties. However, our method also raises the need for further research in key areas where significant uncertainties remain, for example the possibility for non-linearity of the poroelastic response to surface loading and Earth tide forces. Addressing the identified uncertainties could contribute towards improving subsurface monitoring and characterisation in both consolidated and unconsolidated systems."*

We hope that this clarifies what we mean.

Since it is now clear that the uncertainty in parameter estimates is not addressed, this point should be mentioned in the discussion or the conclusions. To mention a very basic point, it is unclear how any measurement error will propagate to computed parameters.
This statement contradicts the earlier comment by the reviewer where they requested omission of uncertainty considerations in the discussion. We believe that we have addressed this in our earlier response.

Other minor points:
At line 120: I guess "programs" should be replaced with codes/tools/software or similar.
Done.

Formatting of equations (6)-(12) should be improved. Remove the "and" between equations (unnecessary).
We believe that equations should be incorporated into the flow of the text for improved readability. Instead of removing the connecting words, we have revised these to improve the meaning and flow of the text.

If possible provide a physical definition for all quantities introduced, or explain the role of each equation (as done neatly in other sections). Same applies to eqs (32)-(35).
We have carefully added physical definitions wherever possible. Note that parameters, for example those related to Kelvin functions, are purely numerical.

Generally I suggest revising all Figure captions, as they contain some typos or inconsistent formatting.
We have double checked and corrected all figure captions.

**Report #2**
**Summary**
I thank the authors for performing major revisions in the structure and text of the manuscript that make it clearer and more pleasant to read. I also thank them for responding to all my remarks with clear answers. This paper is now undoubtedly worth publishing in HESS after the authors perform some minor corrections listed hereafter.
Thank you, we appreciate the review effort.

**Detailed remarks**
p. 5 line 99 equation (1): the definition of the tidal potential V is missing and should be connected to the incomplete sentence "where M2 is the tidal frequency"
We have corrected this.

p. 6 line 107: when computing theoretical tides, for instance with predict.for function within ETERNA software, a model is used for the Earth. This model can be elastic or anelastic. In the later case a phase is

introduced in the tidal constituents. In your tidal prediction using PyGTide, global anelasticity of the Earth should be considered to have correct M2 and S2 theoretical phases. Please provide some additional information on the Earth's model used for the tidal predictions.
We have added the following to the manuscript:

"*This is based on ETERNA and uses the Wahr-Dehant-Zschau model (Wahr, 1981; Dehant and Zschau, 1989) which assumes an elliptical, rotating, inelastic and oceanless Earth.*"

p. 10 line 202: "(e.g., -0.5°C in Figure 3b)" Please remove the "C" since this is degree not degree Celsius.
We have corrected this.

p. 10 lines 202-203: "Figure 2c,d show the solution space when considering the strain response as well as separation of hydraulic properties" This sentence seems unconnected to previous ones and to Fig. 3. Please clarify this remark by adding for instance "at leaky conditions".
We have corrected this.

p. 19 Figure 4: it is strange to see pressure in m instead of Pa. Maybe add in the legend something like "barometric pressures (in equivalent water heights in m)"
This allows direct comparison of the magnitude of influences. We have revised the caption as follows:

"*Time-series of groundwater levels from bores GW075409.1.2 and Thirlmere 2 located at Thirlmere Lakes (NSW, Australia), barometric pressures (in m equivalent water heights for easier comparison to the groundwater pressure heads), corresponding theoretical Earth tides (in nano-strain, nstr) calculated using PyGTide.*"

References: please correct references for repeated url or doi.
We corrected these. Note that it was not possible to highlight this in the track changes. However, we believe that typesetting to standard format will take care of this issue.